# Global sampling decline erodes science potential of natural history collections

Owen Forbes ✉, Andrew G. Young & Peter H. Thrall

The world's natural history collections hold over two billion specimens, representing a unique spatial and taxonomic record of biodiversity on Earth over time. In recent decades, the accessibility and value of collections data have grown through specimen digitisation, enhanced connectivity, and enriched information from new genotyping and digital trait extraction. These advances are expanding the relevance of collections beyond taxonomy and evolutionary biology to fields like environmental monitoring, agriculture, biosecurity, and public health. However, their utility for addressing major global challenges relies on mobilising legacy data and continuing specimen collection and digitisation. Here we show substantial declines in the rates of collection of specimen data over recent decades, from analysis of over 150 million records from the Global Biodiversity Information Facility (GBIF) spanning more than two centuries. The degree and timing of decline varies across taxonomic groups and geographical regions. Overall, these findings suggest that the value of natural history collections as global research infrastructure is eroding due to decreased collecting of specimen data across species, locations, and time. This is occurring precisely when applications for these data have never been more important, and advances in data analytics, AI and genomics promise to unlock deeper insights from natural history collections.

The world's natural history collections contain between 2 and 4 billion preserved specimens of plants, animals and fungi dating back over 400 years[1,2]. Together, these collections represent the most comprehensive taxonomic and spatial record of the Earth's biodiversity and its change through time, which is increasingly driven by accelerating ecological crises.

Over the last 25 years, access to global collections data has improved significantly, owing to increased rates of specimen digitisation and enhanced connectivity via digital data aggregators such as the Atlas of Living Australia (ALA) and the Global Biodiversity Information Facility (GBIF)[3,4]. At the same time, the richness of information associated with specimens continues to grow, with new approaches to genotyping[5], imaging[6] and AI-based trait extraction[7], adding valuable genomic, phenotypic and environmental data layers to previously existing information about specimen identity, location and collection date.

These parallel trends in data richness, analytical tools, and accessibility have transformed the scientific value of individual specimens and extended the utility of natural history collections as research tools beyond traditional use cases in taxonomy and evolutionary biology. Emerging uses of specimen-based data now include a broad and rapidly evolving range of research and management applications. For example, specimen-based observations from the Australian Virtual Herbarium were used to quantify impacts of the 2019–2020 'Black Summer' bushfires on native vascular plant taxa, revealing the vulnerability of these ecosystems to regeneration failure and landscape-scale decline[8]. Collections data have also been central to documenting declines in insect populations, as demonstrated by studies using entomological collections to track substantial losses in abundance and distribution of pollinators such as bees and butterflies over time[9]. Specimen records of bats have been analysed to assess spillover risk of

National Research Collections Australia, CSIRO, Building 803, Clunies Ross St, Acton, ACT, Australia. ✉e-mail: owen.forbes@csiro.au

bat SARS-related coronaviruses in Southeast Asia, providing valuable insights for targeting surveillance and prevention programs for zoonotic disease overflow and epidemic prevention[10]. Fish specimens from museum collections have been used to evaluate historical baselines of marine biodiversity, helping to assess the impacts of overfishing and climate change on ocean ecosystems[11].

These examples illustrate that the increased richness, availability and connectivity of specimen-based data are repositioning natural history collections as globally relevant research infrastructure. However, this expanded value is fundamentally dependent on their taxonomic breadth, geographical range and ongoing temporal coverage. This means that the utility of natural history collections for understanding and responding to the world's major biological and environmental challenges depends both on the ongoing mobilisation of legacy specimen data, as well as the continued collection, characterisation and digitisation of new specimens from across geographical regions and clades of the tree of life. Indeed, our ability to effectively leverage collections data for studying and responding to environmental and societal challenges depends on ongoing specimen collection of sufficient volume, taxonomic breadth and spatial coverage to provide the information needed to understand historical and contemporary patterns and predict future trends.

While ecological data from field observations, citizen science and remote sensing continue to grow, specimen-based records remain crucial as they enable accurate species identification, verification and quality assurance for other data streams, and they possess unique historical extent and temporal continuity. In addition to these essential functions of ground-truthing and verification, as Nachman et al. highlight, specimens are essential for discovering new species, tracking environmental degradation, studying morphology and physiology, investigating gene expression and epigenetic modifications associated with environmental adaptation, and extracting novel information as analytical technologies continue to advance[12,13]. This capacity for repeated examination and data extraction makes specimen collections an irreplaceable scientific resource with unique value beyond what observational data alone can provide.

Given the critical dependency on spatiotemporal representation of global patterns in biodiversity underpinned by collecting activities, it is important to explore trends in how herbaria and museums are adding specimens, representing diverse species, geographical locations and environments, and how these collecting trends are changing over time. Previous studies have identified patterns of decline in specimen collection across specific taxonomic groups and regions. For example, Prather et al. documented a significant decline in plant collecting in the United States[14], while Malaney and Cook highlighted reduced availability of mammalian specimens as an issue of particular concern during the present era of rapid environmental change[15]. More recently, Rohwer et al. demonstrated declining growth in vertebrate collections globally using GBIF data[16]. Building on these important contributions, we conduct a comprehensive global analysis across major taxonomic groups. We analyse over 150 million biological specimen records aggregated by GBIF covering the period 1800 to 2024, representing three major taxonomic groups of Chordata, Plantae and Arthropoda. Specifically, we ask three questions: (1) How has the number of specimens incorporated annually into collections changed, particularly over the last 70 years? (2) What change has there been in the number of unique species that these specimens represent? (3) What change has there been in the geographical extent and distribution of specimens being sampled?

## Results
Analysis of GBIF records shows that there have been significant declines in specimen data added annually over the last several decades. However, the degree and timing of decline vary among taxonomic groups and across geographical regions.

## Trends in specimen counts, unique species and spatial extent
Figure 1 presents results for the number of new specimens collected per year, number of unique species per year, and spatial extent measured by the number of 1-degree grids with specimens collected per year, across three major taxonomic groups: Chordata, Arthropoda and Plantae. After filtering for records with valid entries for species and year of collection between 1800 and 2024 and additional data cleansing to remove several duplicated and anomalous datasets, the final aggregated data used in analyses included: 26,632,273 specimens for Chordata (16,898,122 or 63.4% with associated location data)[17]; 87,490,390 total specimens for Plantae (51,346,527 records or 58.7% with associated location data)[18]; and 45,704,719 total specimens for Arthropoda (36,954,262 or 80.8% with associated location data)[19]. Here we focus on the findings for the period 1950–2019, representing the majority of collecting activity over the last two centuries, noting that the full results for 1800–2024 are available in Supplementary Fig. 1. As described in the Methods, we focus on collection years prior to 2020 due to uncertainty around the impacts of the COVID-19 pandemic, and to avoid over-interpretation of trends in very recent years with unknown accession and databasing delays.

For Chordata (Fig. 1, left column), declines in specimens per year, unique species and spatial extent are apparent from 1966 onwards, with an acceleration in decline around 2010. Between 2010 and 2019, the average number of Chordata specimens per year was 206,588, representing a 47.0% reduction relative to the peak 5-year moving average of 389,900 specimens per year in the period 1964–1968. We also investigated trends for specimens within more specific classes for birds, mammals and fish, and found similar patterns overall with a slight trend of less pronounced and more recent decline for birds, compared to a long-term steady decline since the 1960s for mammals and fish (Supplementary Fig. 14).

For Plantae (Fig. 1, middle column), the 5-year moving average for specimen-based records peaked between 1980–1984 but declined from 1985 onwards, with an acceleration in that decline from 2005 in number of specimens collected, number of unique species, and spatial extent of specimens collected. Between 2010 and 2019, the average number of Plantae specimens per year was 661,760, representing a 43.0% reduction relative to the peak of 1,161,296 specimens per year in the period 1980–1984.

For both Chordata and Plantae, the increase in overall spatial extent between 1990 and 2005 coincides with a period of declining overall number of specimens, indicating increased collection of specimens from under-represented areas that historically had lower collecting activity, but with sharper declines in areas that have traditionally had higher collecting activity. For instance, during this period, Chordata collecting grew in Asia and South America, while declining in North America, Europe and Oceania (Supplementary Fig. 15). Similarly for Plantae between 1990 and 2005, collecting activity increased in Africa, Europe and South America while decreasing in Oceania and North America (Supplementary Fig. 16). Detailed spatial trends are explored further below.

For Arthropoda (Fig. 1, right column), the trends differ somewhat from Chordata and Plantae, with a more recent peak and the beginning of decline starting from 2010 for specimens per year and spatial extent. The number of unique species per year for Arthropoda demonstrated moderate growth from 1950, until a decline began around 2005. As we address in the Discussion, Arthropoda may demonstrate differing overall collection trends and may be more substantially impacted by backlog issues and database accession delays than other groups. Between 2015 and 2019, the average number of Arthropoda specimens collected per year was 672,344, representing a 27.3% reduction relative to the peak 5-year moving average of 924,910 specimens per year in the period 2008–2012.

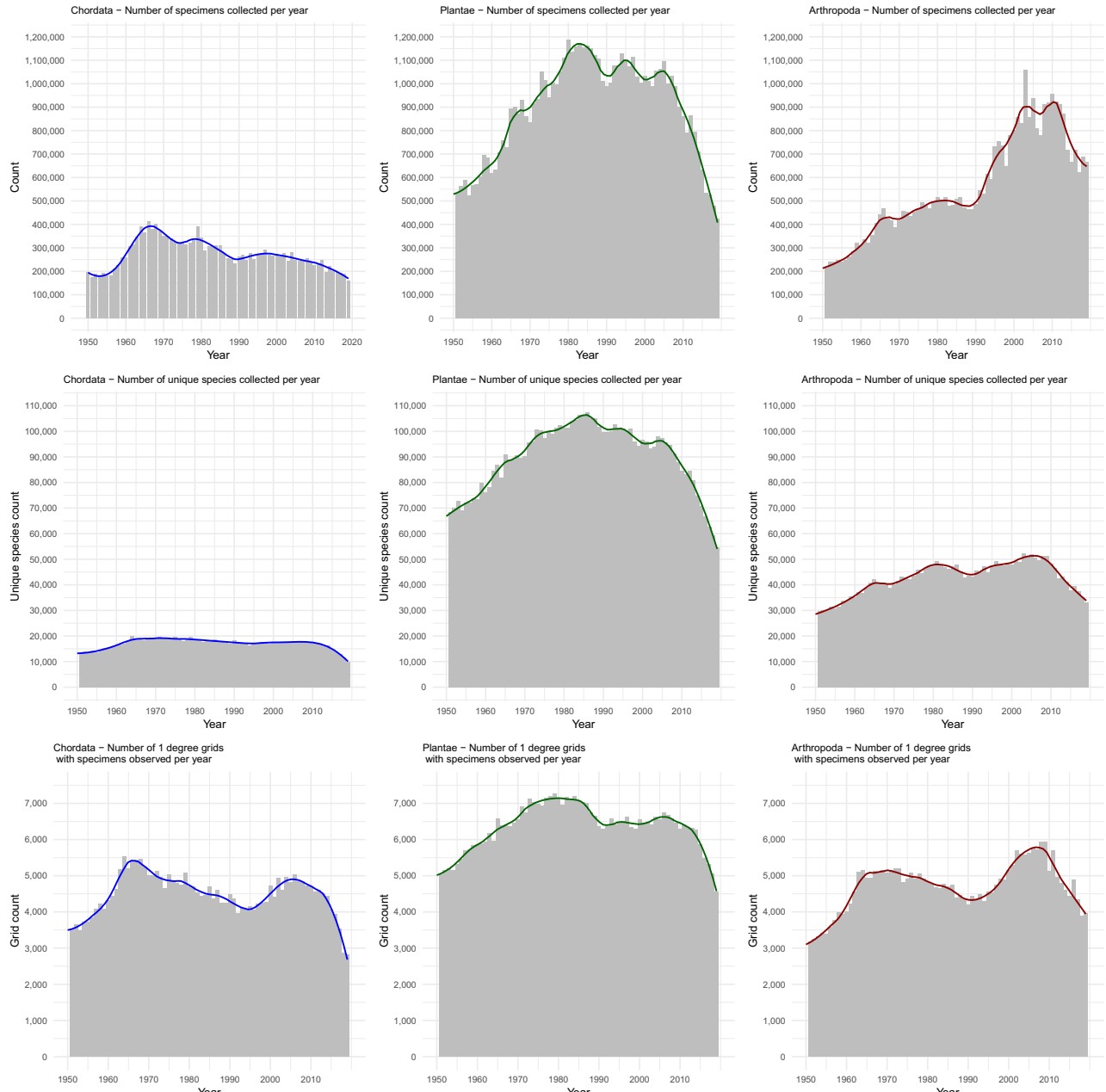

**Fig. 1 | GBIF specimen records–number of specimens, unique species and spatial extent for Chordata, Arthropoda and Plantae (1950–2019).** Left column = Chordata; middle column = Plantae; right column = Arthropoda. Top row = number of specimens per year; middle row = number of unique species per year; bottom row = spatial extent based on the number of 1-degree grid cells with specimens observed per year. Blue, green and red lines indicate LOESS (locally estimated scatterplot smoothing) curves for specimen counts by collection year for Chordata, Plantae and Arthropoda, respectively.

## Forecasting

We implemented forecasting analyses using autoregressive integrated moving average (ARIMA) models based on GBIF historical database snapshots between 2007 and 2023, to assess how trends over collection years are likely to evolve into the future as more specimens are databased and more institutions contribute their datasets to GBIF. ARIMA models are particularly well-suited for this analysis as they account for temporal dependencies in time series data while allowing for non-stationary trends[20]. In the absence of accession dates for GBIF records, these models provide a robust alternative approach to understanding the timeline between specimen collection and their appearance in global databases. Each ARIMA model captures the historical pattern of how records for a specific collection year have accumulated in GBIF over time, driven by

processes including new institutions sharing their collections to GBIF, and the overall timeline from new specimen collection, local taxonomic identification, and institutional databasing, through to eventual GBIF accession. Forecast models were estimated for the collection years 1950–2019. Figure 2 presents forecasting results for specimen counts by collection year, at 5-year (2028) and 10-year (2033) horizons from the December 2023 database snapshot. Overall, these ARIMA forecasts indicate that, based on the historical patterns of growth in GBIF records over the time series of database snapshots between 2007 and 2023, currently observed trends in decline are unlikely to substantially change at 5-year and 10-year forecast horizons, as historical and newly collected specimen records continue to be added. Uncertainty is higher for forecasts in more recent years, where the analyses are based on fewer snapshot

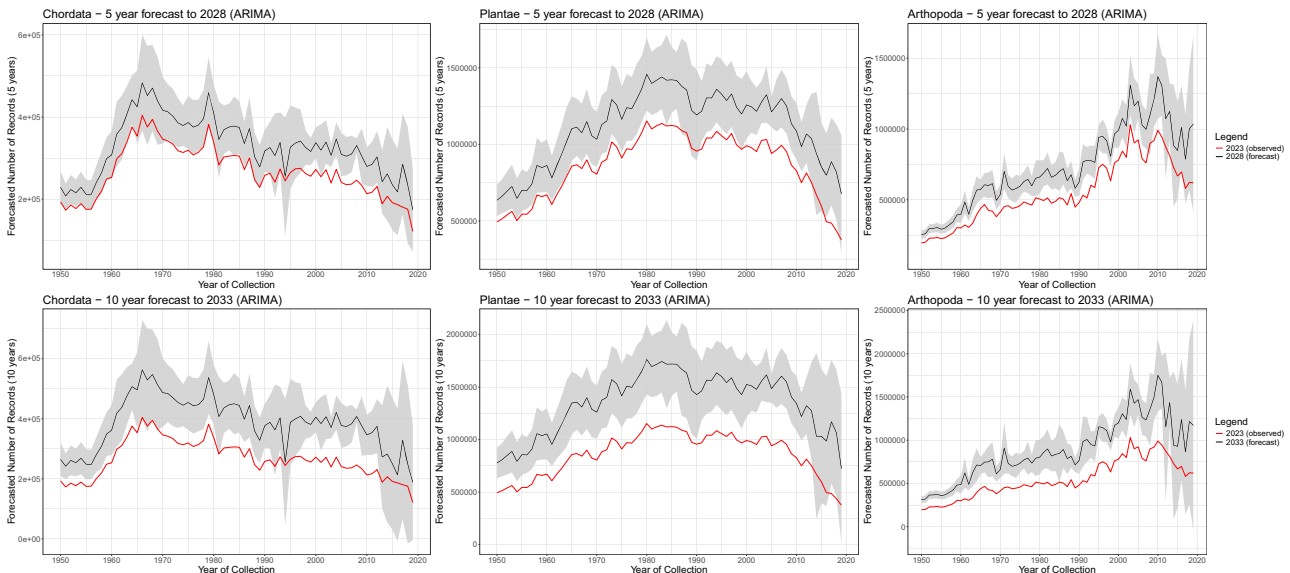

**Fig. 2 | Autoregressive integrated moving average (ARIMA) forecasts for GBIF specimen records—number of specimens by year of collection (1950–2019).** Top row = 5-year forecasts to 2028; Bottom row = 10-year forecasts to 2033; Red lines indicate observed data in 2023; Black lines indicate mean forecast values; Grey ribbons indicate 95% confidence intervals. Forecasts were estimated for the collection years 1950–2019.

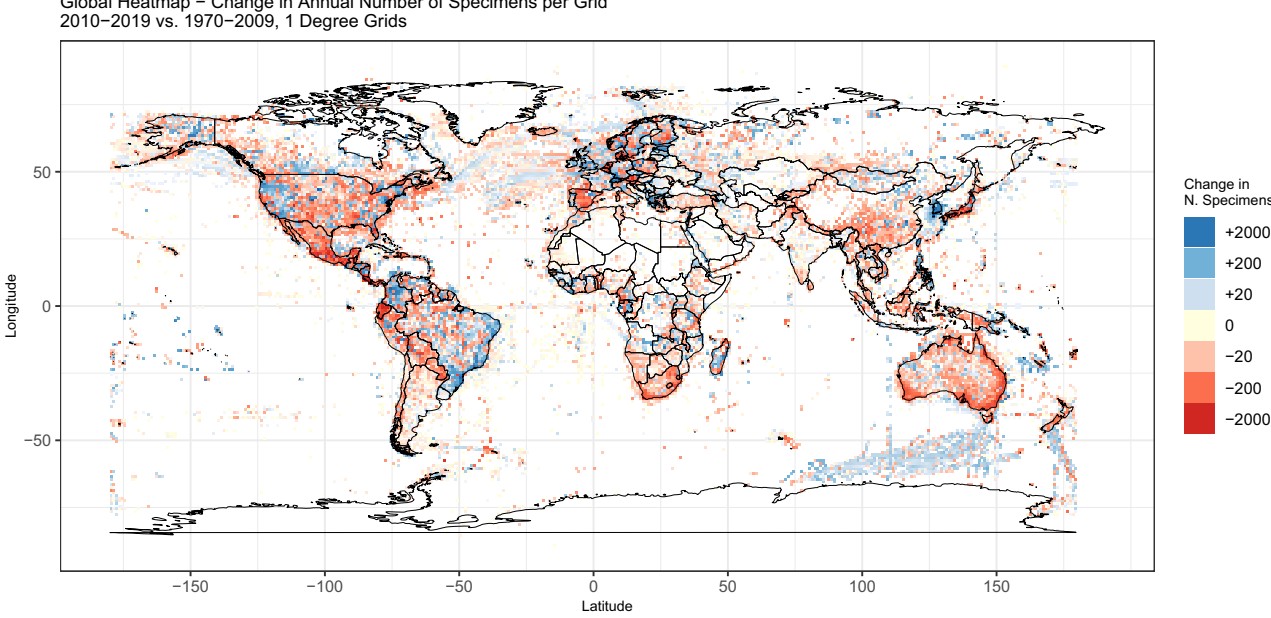

**Fig. 3 | Global heatmap of change in mean annual number of specimens collected, 2010–2019 vs. 1970–2009.** This heatmap displays the change in the mean annual count of specimens between 2010 and 2019 compared to the reference period of 1970–2009, in 1-degree latitude-longitude grids, across all three taxonomic groups combined (Chordata, Plantae and Arthropoda). A diverging colour palette is presented on a log10 scale.

timepoints. In-sample validation results and details on differencing order selection for ARIMA models are provided in Supplementary Note 7.

For Chordata, the decline of 47.0% in 2010–2019 relative to the peak collecting period of 1964–1968 is expected to persist with a 46.0% decline forecast for 2033. For Arthropoda, although specimen-based records peak much more recently and demonstrate faster growth for recent collection years (Supplementary Fig. 12), the decline of 27.3% in 2015–2019 relative to the peak collecting period of 2008–2012 is expected to persist with a 27.8% decline forecast for 2033. For Plantae, the decline of 43.0% in 2010–2019 relative to the peak in 1980–1984 is forecast to reduce, though remaining substantial at a 35.3% decline forecast for 2033.

## Spatial trends

Assessing combined spatial trends for these three taxonomic groups, we found that there is substantial variation in the timing of peaks and declines in collection across continents (Supplementary Fig. 2). More recent peaks in collection are apparent around 2012 for Africa and Asia. Decline is apparent from around 2005 in Europe, North America and South America. A much earlier trend in decline is evident in Oceania from around 1995. Trends by continent within each major taxonomic group are available in Supplementary Figs. 15–17 and explored in the Discussion.

To further investigate spatial trends at a finer geographic scale, Fig. 3 shows a global heatmap of the discrepancy for mean number of specimens annually collected within 1-degree grids, in 2010–2019

relative to the annual average in each grid over the broad peak collecting period from 1970 to 2009. Certain countries, such as South Korea and Brazil, show increased collection in this period relative to preceding decades. There is also an apparent trend of increased marine specimens, for instance, those collected to the south of Australia and between Tasmania and Antarctica. However, the overall trend of substantial decline is evident, particularly in areas with traditionally high concentrations of collections data in North America, Western Europe, South Africa, Japan and Australia.

It is important to note that the geographic distribution of institutions contributing to GBIF is not uniform, with a higher concentration in North America and Europe (Supplementary Fig. 18). This geographic bias in data contribution may influence the observed spatial patterns in specimen collection trends, particularly in regions with fewer contributing institutions. The extent to which these geographic biases influence observed trends in collecting is difficult to quantify.

## Discussion

### Collections data are in decline
Based on our analysis of specimen-based data from GBIF, the rate at which natural history collections are adding specimens is showing a declining trend, having peaked between 1964 and 1968 for Chordata, between 1980 and 1984 for Plantae, and more recently between 2008 and 2012 for Arthropoda, overall diminishing substantially in recent decades. More specifically, across diverse major taxonomic groups (plants, vertebrates and insects), fewer specimens representing a narrower range of species are being collected over a smaller spatial extent.

This decreasing spatiotemporal and taxonomic coverage of global biodiversity represents a significant erosion of one of the primary differentiating values of specimen-based collections. This is particularly concerning given that we face escalating and complex environmental challenges that necessitate the use of specimen-based data to understand, predict, and address challenges related to climate and land-use change, biodiversity decline, deteriorating ecosystem services and ecological collapse[21]. This decline is happening at the very time that the emergence and ongoing development of high-throughput digitisation, novel museum genomics platforms, new data analytic tools, predictive modelling approaches and AI-powered applications are enhancing and extending the ways in which natural history collections can contribute to tackling these problems. While the urgency and intensity of ecological crises are accelerating, advances in statistical data science and machine learning tools are improving our ability to derive actionable insights from these data. However, the spatiotemporal coverage and quantity of collection data, as well as fundamental curatorial and taxonomic expertise, are rapidly declining, leading to impaired ability to address pressing questions, reduced statistical power and higher uncertainty around resulting inferences.

Importantly, based on 5-year and 10-year ARIMA modelling results, the observed trends of long-term declines for Chordata and Plantae specimens are forecast to remain at a similar magnitude and timing. We note, however, that at this stage, it is difficult to confidently forecast the transient effect of the 2020-2025 global pandemic on this trend. Although the peak in collecting activity is more recent for Arthropoda, a substantial decline is still forecast for the period 2015-2019 relative to the peak in collecting activity observed for the years between 2008 and 2012. Overall, while the broad negative trends in collecting activity are consistent across taxonomic groups, more detailed exploration of the data also shows some noteworthy differences.

### Specimen coverage varies by region and taxonomic group
Our results show that overall collecting levels have been historically lower in Asia, Africa, and Oceania than in other geographic regions,

with the highest average number of specimens per year being collected in North America. In 2010-2019 relative to preceding decades, collecting has increased in some areas like Brazil (likely driven by high public and scientific interest in the Amazon basin) or in some marine regions (e.g. between southern Australia and Antarctica, possibly reflecting significant investment by the Australian government in marine research infrastructure. However, large declines in collecting across other biodiverse areas such as Central America, South Africa, and Australia are of particular concern, as are the obvious gaps in biodiversity information for much of northern Africa and the Arctic (Fig. 3).

Considerable variation in collecting patterns exists across taxonomic groups and continents (Supplementary Figs. 15-17). While Plantae and Chordata show similar patterns when aggregated globally, within geographic regions, there are some marked differences. For instance, in Europe, Chordata collecting peaked between 1965 and 1985, while Plantae peaked around 2010. Within Chordata, collecting in both Africa and North America peaked in the 1960s and showed subsequent sharp declines, albeit probably for different reasons. Chordata collections in South America have changed little since the 1960s, with evidence of a decline only in very recent years, while in Asia, collecting may not have peaked yet. Interestingly, there was less variability across sub-groups within Chordata compared to this geographic variation (birds, fish, mammals; Supplementary Fig. 14).

Arthropoda collecting patterns differ from other groups, with peak collecting generally occurring more recently (2005-2015) at the continental scale, except in Oceania (1995-2005). In some regions, collecting activities have continued to increase until very recently, making it difficult to determine if a peak has been reached, or whether there is evidence for a sustained decline (e.g. in Africa, Europe). The complex trends in Arthropoda may be driven more by processing burdens than reduced collecting effort, given the greater diversity, high proportion of undescribed species, and increased handling time for specimens. Regardless of the causes, there is a substantially reduced data scope available to understand spatiotemporal trends in the last decade. This speaks to the value of developing data-driven priorities for collections research, to maximise utility and information gained from limited resources and staff capacity (see 'Future directions').

### Potential drivers of decline in collections data
There are likely multiple causal factors associated with the observed decline in collecting activity (e.g. geopolitical, socioeconomic, support for scientific and digital infrastructure, resources for growing and training capability, shifting public perceptions, etc.), the relative importance of which will vary by country and institution. While analysis of how various drivers may interact at institutional, local, national and global scales to drive observed collecting patterns is beyond the scope of this study, we highlight some key considerations here:

a) Resource constraints: Limited funding sources for collections, coupled with rising labour and infrastructure costs, and stricter fieldwork safety requirements have dramatically increased per-specimen costs and reduced institutional capacity for collecting, processing, and housing specimens[22–24]. For many natural history collections, ever-declining budgets mean that basic maintenance and collection management may be all that is feasible, despite growing demands for specimen-based data.

b) Regulatory and ethical considerations: These include safety/risk assessments (e.g. biosecurity and conservation concerns), which increasingly drive approval processes for collection permits. The rapid growth of alternative data streams (e.g. autonomous sensors, audio recordings, citizen science observations) likely contributes to views that the continued collection of physical specimens needs much stronger scientific justification than would have been the case in the past. This is particularly the case for

vertebrates, where public sensitivity to collecting activities is high, though is not as relevant as a driver to consider for plants. Emerging national and international initiatives aimed at recognising the rights of traditional owners (e.g. the Nagoya Protocol, the Global Indigenous Data Alliance, CARE principles) add further complexities, requiring thoughtful navigation[25–27].

c) Shifting workforce priorities: Growing recognition of the need to 'future-proof' natural history collections is driving capability expansion (e.g. digitisation, genomics, big data analytics and ML/AI applications)[28]. The demand for greater skills diversity in collection workforce roles means stretched funding for traditional curatorial expertise (see (e) below), exacerbated by the fact that, as collecting activities decline, fewer people are trained in collection techniques, curation and basic taxonomy[22].

d) Targeted collecting trends: There has been a shift towards collecting only specific target species rather than holistic, broad-based censuses of species present in an area[29,30]. As we argue below, to some extent, future collecting activities may need to further shift the focus from species discovery (clear exceptions being geographic regions with biodiversity hotspots or hyper-diverse groups such as insects) to characterising and predicting eco-evolutionary processes in a changing world.

e) Funder and public perceptions: There continue to be persistent gaps in public awareness and funding body understanding of the role of natural history collections or how sampling, identification, and preservation of physical specimens critically underpins the ability of collections to contribute to addressing environmental problems[31]. Importantly, collections-based work by formally trained taxonomists provides the essential knowledge base for other data sources such as long-term ecological sites, citizen science initiatives, and indeed any aspect of biodiversity science[32].

f) Species discovery: Related to points (d) and (e), is the possibility that observed declines in rates of species discovery (especially for Chordata and Plantae, which are much less diverse than Arthropoda) are related to these taxonomic groups approaching higher levels of taxonomic saturation (e.g. in Europe and North America)[33]. This could negatively impact the perceived need for continued collecting and thus the levels of institutional and funder support.

## Potential impacts on collection science

The unique value of natural history collections is based on comprehensive spatiotemporal coverage of biodiversity at local, regional, continental and global scales. The over-arching consequence of the observed decline in collecting activities is to significantly erode this value, hampering advances in biodiversity science, and ultimately reducing the ability of collections to assist with understanding and responding to global crises such as climate change and biodiversity loss. This decline is particularly concerning for evidence-based policymaking, conservation, management of ecosystem services and crisis management, which rely on accurate predictions of spatial and temporal trends. Decreased specimen data coverage leads to lower accuracy and higher uncertainty in statistical inferences from collection data, and also undermines the utility of other ecological data streams, which are dependent on accurate species identification. These trends are compromising our ability to estimate ongoing climate and land-use change impacts, model species distribution trends, and predict future ecological and evolutionary impacts. This data gap further increases the risk that we will be unable to adequately manage or protect the ecosystems that represent our greatest natural resource, providing critical life support systems for humanity and Earth's biosphere.

Importantly, for natural history collections to mobilise, enhance and add value to the fundamental specimen data they hold, it is critical that they also deploy and integrate emerging technologies. This includes developing approaches to link taxonomic, geolocation, environmental, phenotypic and genetic data layers, and building predictive models and analytical tools to fully leverage biodiversity data. While increasingly, collections have new data science tools and technical platforms at their disposal, the trend of decreasing growth in collection data inhibits their capacity to leverage these tools to their full potential, leading to underutilisation of such technological advances, and decreased ability of collection science to contribute to global initiatives. Arguably, this is happening at a historical environmental juncture when these unique scientific contributions are most urgently required[21].

## Future directions

As a scientific discipline, we need to take action to address this problem. To fully realise the future scientific value and impact of natural history collections, we must first look critically at their role going forward, not in isolation but as part of a dynamically evolving biodiversity information ecosystem. This ecosystem is being rapidly populated by a diverse range of emerging data streams, including citizen science observations, satellite and remote sensing information, eco-acoustics and environmental DNA. Each of these delivers valuable data but is associated with its own biases and errors. Focusing on the key attributes of specimen-based collections (spatial coverage, temporal depth and taxonomic certainty and breadth) provides the basis for assessment of what information specimens can uniquely deliver as well as their complementarity to these other data sources. This will be achieved most usefully when framed by the specific data needs of emerging application domains. This involves considering what types of data are most valuable for specific research questions, how the spatiotemporal scale relates to different research domains (e.g. species discovery, eco-evolutionary processes, climate and land-use change impacts), and which analytical tools can maximise the impact-to-effort ratio in research prioritisation. Importantly, these considerations apply not only to new specimen collection but also to increasing and optimising the databasing and digital mobilisation of existing specimen backlogs, particularly for hyper-diverse groups like Arthropoda, and to choosing which data layers (e.g. genomic, phenotypic and environmental) are most valuable to add.

We encourage adoption of data-driven approaches to specimen collection and digitisation that optimise the impact-to-effort ratio in addressing urgent environmental challenges. Decisions on where to focus collection efforts to accelerate data streams should be based on current and future trends, some of which have recently been explored[34]. For example, with respect to future collecting activities, where should we sample? What should we sample? How much should we sample, and at what spatial and temporal scales?

A strategy for 'collecting smarter' could respond to resource constraints by implementing systematic methods for prioritising projects. To guide future collecting activities and data mobilisation efforts, we envision interactive platforms incorporating Bayesian decision theory and optimal design tools to support collection sampling design. Using metrics such as Value of Information or expected information gain, these platforms could help determine optimal data-driven sampling strategies, coordinate strategic sampling at regional and larger scales, and ensure collections-based biodiversity data is maximally useful for various applications such as climate change research and biodiversity loss assessment.

The scientific community as a whole needs to collaboratively advocate for the value of digitised and mobilised collection data in tackling global environmental problems. While collection scientists bring crucial expertise, the growing applications of specimen data across diverse disciplines necessitate broader engagement from researchers in ecology, genomics, climate science and other fields. Building awareness of the tremendous value inherent in natural history collections, both in the public (e.g. via citizen science initiatives) and in

funding bodies, is crucial for securing resources needed to grow collections and address structural and capability constraints. Similarly, structured exploration of the data needs of relevant research disciplines and applications will help to shape what collections data should most usefully look like in the future. Collections should adopt a forward-looking mission that emphasises eco-evolutionary processes and predictive tools, while maintaining focus on fundamental species discovery in biodiversity hotspots and understudied areas. This shift could reinvigorate interest in collecting activities, even in regions approaching taxonomic saturation. At the same time, it will be important to help upskill and support institutions in less developed countries to be part of this shift, which may require additional digital infrastructure and new capabilities.

Developing and extending data infrastructure is crucial for harmonising collections-based data with other ecological and environmental data sources, responding to declining collecting trends by maximising the value of each collected specimen. Existing examples of successful implementation of novel digital infrastructure include the data architecture standards developed by international platforms such as GBIF, DiSSCo and iDigBio[35]. These technological innovations have spurred advances in collections science by enabling the implementation of the digital extended specimen concept, transforming static records into dynamic, interconnected knowledge systems that integrate traditional specimen data with derived information like genomic sequences and phenotypic traits, alongside other ecological data layers.

Expanding the reach of collections may also require integrating specimen metadata with other data sources (e.g., citizen science, remote sensing), while leveraging AI-based tools to improve the quality of public observations and address data quality-quantity trade-offs. While taxonomic expertise remains essential for accurate species identification, AI-based tools can significantly enhance species identification accuracy and consistency across various data types and quality standards. Paradoxically, the decline in collecting activities threatens to limit the training data needed for AI/ML tools, potentially impeding their development and effectiveness.

## Caveats

While our analysis provides compelling evidence for a decline in specimen-based data across major taxonomic groups, several caveats warrant acknowledgment. GBIF data represent only a subset of global natural history collections, as participation is voluntary and most institutions have incomplete digital coverage. Our study specifically filtered for preserved specimens (~10% of GBIF's total holdings) to focus on specimen collection trends rather than broader biodiversity observation patterns. A significant data limitation and analytical challenge is the absence of accession dates in GBIF records. Without these dates, we cannot fully disentangle three distinct processes: (1) actual declines in specimen collecting activity, (2) time lags between collection and local databasing, and (3) delays between local databasing and GBIF contribution. To address this data limitation, we implemented forecasting analyses using historical GBIF database snapshots. See Supplementary Notes 5, 6 for a detailed discussion of these data challenges.

Despite these caveats, the patterns we have identified are consistent across major taxonomic groups and geographic regions. Our study, based on over 150 million datapoints from numerous collections worldwide, provides a robust foundation for our conclusions. While acknowledging the inherent limitations around data quality and comprehensive cleaning for aggregated biodiversity data[36], the overall patterns in our study remain substantial. The temporal consistency and taxonomic breadth of our findings, coupled with ARIMA forecasting results, suggest that the observed declines in specimen collecting represent a real phenomenon rather than an artefact of data management processes. These findings underscore the urgent need

for continued investment in and strategic planning for natural history collections to ensure their vital role in addressing global environmental challenges.

## Concluding remarks

In an era marked by rapid climate change and biodiversity loss, it is imperative that we continue to invest in the unique value of natural history collections data. We argue that such resources not only directly contribute to addressing the increasingly urgent need for actionable ecological information but also underpin other critical biodiversity and environmental data sources (e.g. citizen observations, remote sensing, genetic information) that depend on accurate species identification. Thus, the decline in collections data is a worrying trend that requires strategic approaches to how we resource and target ongoing collection efforts, to ensure that we can effectively meet the accelerating environmental challenges of our time.

To be explicit, we are not arguing for a return to the same levels of collecting activities that occurred in the past, but rather that we need to collect 'smarter', especially given ongoing resource constraints. Modern collecting practices need to reflect the expanding scope of applications for specimen data beyond traditional taxonomy, and embrace the diversity of information types that can now be extracted from each specimen through advances in digitisation, genomics and AI-powered trait extraction. In practical terms, this might mean objectively evaluating whether additional physical specimens are required for a particular research or application objective, or whether existing data might suffice. This determination is becoming increasingly important given the rapidly growing body of information from other data streams, including citizen science and field observations, that now make up ~90% of observations in GBIF, alongside eDNA and remote sensing data. If new collecting activities are indicated, then sampling should aim to maximise the value of the additional information gained. This can be achieved in part, via the development and deployment of tools that aid in effective integration across data layers, and the application of data-driven sampling design frameworks in the context of this broader data ecosystem.

We are only starting to fully appreciate the value of the irreplaceable data held in natural history research collections at a time when their unique spatiotemporal coverage is under threat. It is crucial that we work together to strengthen, protect and invest in specimen-based data collection for the sake of science, humanity and the planet. It is worth noting that while, to some extent, we can recapture spatial information about species occurrences, we cannot recapture time. And time is something that our natural systems do not have.

## Methods

We analysed large-scale data from the Global Biodiversity Information Facility (GBIF) for specimen-based records of Arthropoda, Chordata and Plantae[17–19]. For each taxon, we exported all specimen-based records from GBIF, and initially filtered the data to keep only those records with valid, non-missing entries for species and year of collection between 1800 and 2024. For spatial analyses, we further filtered each dataset to only include records with valid, non-missing entries for decimal latitude and longitude. For each taxonomic group, we calculated the number of records per collection year, the number of unique species represented per year, and the number of 1-degree grids with records present per year, as a measure of spatial extent.

### Data cleaning

Through visual inspection of plots for contemporary records and historical snapshots of the GBIF database, we identified a small number of anomalous instances where the numbers of specimens recorded for an individual institution in certain years of collection were so disproportionately large relative to neighbouring years from the same institution, and also when compared to other institutions, that their

reliability and plausibility came into question. As our goal was to understand overall trends in global collecting activities across these taxa, and these anomalies represented implausible data that impaired our ability to analyse high-level trends and inferences, we chose to implement an anomaly filtering process.

In our analysis, we employed conservative thresholds to identify and remove a small number of anomalous datasets while maximising the overall data retained. We chose thresholds based on standard deviations (SDs) on the log scale compared to the counts in that collection year from other institutions, and also relative to counts in all collection years for that institution. The use of standard deviations on the log scale to detect outliers is a common practice rooted in statistical theory[37,38]. This was designed as a cautious approach, and thresholds were explicitly chosen through iterative testing to remove a small fraction of records that seemed clearly anomalous and implausible, while retaining as much data as possible to support our main analyses. As addressed in the Discussion, this was not designed or intended to be an exhaustive process, and there are likely other instances of data quality issues remaining to be resolved at the institutional or aggregator level. This includes GBIF's ongoing quality assurance processes, which are already in place, as evident in the instances of records being removed at several junctures over time, seen from the historical GBIF database snapshots (Supplementary Figs. 10–12).

The steps involved in this anomaly filtering process were as follows:

1. Log transformation: We applied a log transformation to the yearly counts to stabilise variance, given large variation on the linear scale, allowing similar scale thresholds to be applied across taxa.
2. Standard deviation calculation: For each institution, we calculated the mean and standard deviation of the log-transformed counts.
3. Threshold application: For contemporary records, institution-level datasets that exceeded three standard deviations above the mean log-transformed count (both relative to all years for that institution, and relative to all other institutions in that collection year) were flagged as anomalous. This corresponded to a range of 16.9–104.6 linear scale standard deviations above the mean annual count across institutions, representing an extremely large departure from the typical range. Based on iterative testing, more conservative thresholds of 3.1 log SDs (Plantae and Chordata) and 3.6 log SDs (Arthropoda) were used for anomaly removal in historical snapshots, due to greater variability in the magnitude of institution-level annual record counts in the historical snapshots data.
4. Comparison across years and institutions: These flagged datasets were further inspected and compared against counts from other collection years and other institutions to assess plausibility and reliability.
5. Removal of anomalous records: Finally, records identified as anomalous through these stringent criteria were removed from the dataset to maintain data integrity, and we contacted the contributing institutions for these datasets to identify possible causes for the data anomalies. Of the six institutions contacted, one responded (INSDC EMBL-EBI), who confirmed that their uploads to GBIF duplicated entries from other contributing institutions, including CBG Guelph, another institution whose records were also flagged as anomalous—for Arthropoda data in Canada between 2012 and 2015.

Further details on data cleaning are provided in the Supplementary Notes 3, 6.

## Primary analyses

Analyses were conducted using the open source statistical software R version 4.3.2[39], and packages tidyverse v. 2.0.0[40], sp v. 2.1-3[41,42], sf v. 1.0-15[43,44], spdep v. 1.3-3[45], rnaturalearth v. 1.0.1[46], countrycode v. 1.6.0[47], readr v. 2.1.5[48], tseries v. 0.10-55[49], lubridate v. 1.9.3[50], data.table v. 1.15.4[51], zoo v. 1.8-12[52], scales v. 1.3.0[53], purrr v. 1.0.2[54], arrow v. 15.0.2.9000[55] and forecast v. 8.22.0[20,56]. Visualisations were generated using ggplot2 v. 3.4[57], viridis v. 0.6.5[58], ggpubr v. 0.6.0[59] and gridExtra v. 2.3[60]. All code used for analyses in this manuscript is publicly available on the associated Zenodo repository[61].

For our primary analyses and forecasting (Figs. 1, 2), we restricted data to pre-2020 for two key reasons. First, recent collection years have inherently greater uncertainty due to a shorter elapsed time with unknown databasing and accession delays, making forecasting analyses and inferences for long-term trends based on these years less reliable. Second, the COVID-19 pandemic (2020 onwards) severely disrupted collection operations through cancelled expeditions, reduced staffing, and facility closures, exacerbating specimen backlogs. By focusing on pre-pandemic data, we avoid conflating long-term trends with these extraordinary disruptions while ensuring more robust analyses. Supplementary Fig. 1 includes results up to 2024 for the primary analyses.

Figure 2 (1950–2019) and Supplementary Fig. 1 (1800–2024) present plots showing the number of specimens, unique species, and 1-degree grids (spatial extent) per year, for each of these taxonomic groups after data cleaning.

Combining records across these three taxonomic groups, Supplementary Fig. 2 shows spatial trends in the number of specimens collected over time by continent. Examining spatial trends in greater detail, Fig. 3 presents a global heatmap of change in the mean number of specimens collected per year in 1-degree grids, contrasting recent years (2010–2019) with the mean annual record counts across a period of higher collecting in preceding decades (1970–2009). Further details on spatial trends and splits by smaller taxonomic groups are provided in Supplementary Figs. 14–17.

## Forecasting

These forecasting analyses aimed to assess how temporal trends and the timing of decline in specimens by collection year may change in future, as new records and datasets continue to be added to GBIF. We fit a separate autoregressive integrated moving average (ARIMA) model for each time series of record counts within a collection year, using historical GBIF database snapshots over 17 years (2007–2023), which were provided by GBIF staff and are published alongside this article—see the Data availability statement. We fit ARIMA models for each collection year from 1950 to 2019.

ARIMA is a widely used and well-validated method for time series forecasting[20]. To assess the validity and performance of the ARIMA forecasts, we conducted in-sample validation by using the fitted models to forecast results forward from 2007 at 5-year (2012), 10-year (2017) and 15-year (2022) horizons. As part of this in-sample validation process, we assessed the performance of ARIMA models using zero-order, first-order and second-order differencing, comparing the root mean squared prediction error (RMSPE) between observed and predicted values in these snapshot years. This validation process indicated that for all three taxonomic groups, on average, the best RMSPE was achieved with first-order differencing (Supplementary Table 2). In order to produce comparable forecasts across different collection years and to avoid introducing additional variance through over-differencing, we ran ARIMA models with first-order differencing ($d = 1$). Zero seasonal differencing was used ($D = 0$), as there was no cyclical or seasonal structure in these time series. Autoregressive order (p) and moving average (q) parameters were chosen automatically for each model based on the optimal corrected Akaike Information Criterion (AIC$_c$) values with the function forecast::auto.arima()[20,56]. Detailed forecast validation results are presented in the Supplementary Table 2 and Supplementary Fig. 13.

## Reporting summary

Further information on research design is available in the Nature Portfolio Reporting Summary linked to this article.

## Data availability

All data used for analyses in this work are publicly available. There are no restrictions on data availability. Given the large file sizes of the specimen record datasets, instead of uploading to the Zenodo repository, we direct users to GBIF directly to access the specimen record datasets. GBIF data exports are available at the following links[17–19]: GBIF.Org User. Occurrence Download−Chordata. https://www.gbif.org/occurrence/download/0016915-240425142415019 (2024) https://doi.org/10.15468/DL.CNRXZT. GBIF.Org User. Occurrence Download−Plantae. https://www.gbif.org/occurrence/download/0016914-240425142415019 (2024) https://doi.org/10.15468/DL.8MX5ZK. GBIF.Org User. Occurrence Download−Arthropoda. https://www.gbif.org/occurrence/download/0016913-240425142415019 (2024) https://doi.org/10.15468/DL.23TSXM. The datasets of GBIF historical database snapshots, used for forecasting analyses, as well as other output datasets used for generating figures, are available in the Zenodo repository with the identifier https://doi.org/10.5281/zenodo.14010665[61].

## Code availability

All code files for analyses in this manuscript are available in the Zenodo repository with the identifier 10.5281/zenodo.14010665[61]. This repository contains analysis scripts, data processing pipelines, and documentation. To run the analyses, we recommend opening the Quarto '.qmd' files using RStudio, which is freely available and open source IDE software. A README file in the repository provides instructions for reproducing the analyses.

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

## Acknowledgements

We gratefully acknowledge the contributions of John Waller, who provided access to GBIF historical database snapshots, and offered guidance on approaches to forecasting and accounting for databasing backlogs. We also acknowledge the contributions of Brendan Lepschi, who contributed ideas around drivers of decline in collecting.

## Author contributions

O.F.—Roles: Conceptualisation, formal analysis, investigation, methodology, software, validation, visualisation, writing—original draft and writing—review & editing. A.G.Y. and P.H.T.—Roles: Conceptualisation, investigation, methodology, project administration, resources, supervision, validation, visualisation, writing—original draft and writing—review & editing.

## Competing interests

The authors declare no competing interests.
