## [Peer Review file · Nature Communications]

Global Sampling Decline Erodes Science Potential of Natural History Collections

Corresponding Author: Dr Owen Forbes

Version 0:

Reviewer comments:

Reviewer #1

(Remarks to the Author)

Forbes and colleagues are concerned that the rate of collections of biological specimens are declining across the world and across all taxonomic groups, just at a moment when new technologies in genomic analyses, imaging, and AI are enabling the extraction of data that is uniquely available from preserved specimens at scales (numbers of records) previously unthinkable. Their analysis of the records of preserved-specimens contained in the Global Biodiversity Information Facility (GBIF) database suggests just that. The global peak in specimen collection considered as unique species records per collection-year for vertebrates (1960s), plants (1980s) and arthropods (~2010) are all past, and the spatial spread of those observations is also declining, although the trace of a concerted effort to prioritise undersampled areas results in a resurgence and delayed decline when the records are considered in spatial terms. That latter is an interesting addition to a simple consideration of the fate of collections from a collection-year only perspective.

The patterns they find globally seem to agree with those previously published with GBIF vertebrate data across by Rohwer et al (2022, <https://doi.org/10.1371/journal.pbio.3001613>), Malaney and Cook 2018 for mammal collections in the US (2018, <https://doi.org/10.1093/jmammal/gyy082>), and Prather et al for herbaria records in US (2004, [<https://www.jstor.org/stable/25063934>](<https://research.ebsco.com/linkprocessor/v2-external?opid=xpptz&recordId=tqryfze4db&url=https%3A//www.jstor.org/stable/25063934>)). Forbes et al's work is clearly more comprehensive in geographic, temporal and taxonomic breadth than the three articles I mention, and the use of additional response variable formulations including geographic coverage per broad taxonomic group in this paper make it a more interesting exploration of the data trends. Nonetheless, this work would be improved by acknowledging existing literature that is surely not limited to the three I found above with a quick search. The GBIF platform itself provides some simple graphical analysis of its holdings that readers could be referred to (<https://www.gbif.org/analytics/global>).

The authors start by referencing all the museums and herbaria of the world, and then switch to GBIF, almost as though it was effectively the same thing (accession aside). Do all institutions submit their records to GBIF? Does ANIC? Maybe I should know the answer to the former but I don't, so I think the authors should clarify for the sake of readers.

I suspect that the average reader is likely to mis-interpret the findings presented here, failing to pick up on the authors' subtle framing and deft avoidance of mentioning observation data, including that from citizen scientists, which account for about 90% of GBIF's records. I think they should insert a sentence that explains what GBIF is and where preserved specimens fit in amongst the types of records held there, in the methods if not earlier.

GBIF is used as the source data for this study presumably because of its global reach and the enormous effort already invested by contributors and curators in standardisation. The downside as they acknowledge is that they don't know about accession dates (how on earth is that not a mandatory field?), so no stock-and-flow kind of model can be built that would allow estimation of bottlenecks applicable to different taxonomic groups, and in different country or regional settings. I think the authors should clarify what their assumptions are that supply process. They warn that longer delays between collection and accession are expected for arthropods but a clear statement about what they assume is the case for each taxonomic group is probably warranted so that readers can keep that in mind when considering the results. For Arthropods, since they are attached to the Australian National Insect Collection, perhaps they could use it as an example and describe the proportion of records ANIC holds that are in GBIF? I think the authors are correct in their assumption about declining collection, but without accession dates the dataset is something of a black box within which declining collection and

accession remain rival explanations, no? I think the authors should do what they can to enlighten readers about what the dataset contains and what it is missing - if there is no literature or way to do that then that is an interesting additional finding. [Footnote, my internet search on the topic just found a forum question on GBIF site that could well be from the lead author "of2" - so they have clearly tried to untangle accession and collection - now they just need to communicate the finding in the main paper as it is something that all readers should take away without having to dig in the supplements.]

The authors combine raw data plots with autoregressive integrated moving average (ARIMA) models to project the trend forward in time, which seems appropriate. If accession dates were available then perhaps a hierarchical model separating collection and accession could have been considered. The analysis performed is well documented in the article and at first blush the code required to reproduce the analyses seems similarly well documented. It is presented in way that would be instructive to hypothetical colleagues. Considerable effort has gone into harmonising data and removing potentially artifactual elements.

The plots of temporal trends in collections-by-year in Figure 1 are simple but compelling. I trust they are backed up by the forecasting analysis, handsomely presented in Fig 2 (though the X-axis must be mis-labelled as they end at 2020?) that also suggest ongoing decline, amidst considerable uncertainty. I think the authors should choose either the 5 year or the 10 year forecasts, placing the other in Suppl if desired.

I think the discussion and main claims are problematic at present because the authors don't mention observational data, including but not limited to structured citizen science data, until late in the Discussion (L398). Those data are (for better and worse) far more numerous, and consequently more commonly employed in many of the domains that the authors claim are at risk from declining collections, e.g. from L339-340, "estimate ongoing climate and land-use change impacts, model species distribution trends, and predict future ecological ... impacts [all for estimating spatial and temporal trends to support evidence based decisions on conservation, ecosystem services and crisis management]. While the authors have cited a couple of published examples where specimen data were used for those purposes, one could find thousands where those objectives were being pursued using occurrence data that didn't relate to a collected specimen. The authors need to either address this head on with a critique of the bulk of that activity, making the case for the need to step up specimen collection, or they need to substantially reduce the scope of the claims to those for which physical collections are fundamental.

The future directions section is strong, and could easily include consideration of how physical and observational or remotely captured / sensed data could be combined, or how specimen collection might respond to the far more numerous observational datasets.

L21 Consider 'Our analysis of over X records from GBIF spanning more than two centuries suggests (significant) declines in the **rate of collection of** specimen data over recent decades.'

L25-26 You don't show "reduced availability" right, data are not being removed from GBIF and accession continues, increasing accessibility (L37). I have understood your finding to be that the source (collecting activity) has been dramatically constricted and that signal is appreciable despite an unknown but generally constant effect of variable accession. Or do you think that commitment to accession is declining, in which case the article would need to be substantially modified to indicate that in the end you still have two important rival explanations, both of which are concerning but have different implications.

L29 Consider using specimens for natural history, now that you have linked the two ideas in the first sentence.

L31 Here instead of natural history you are using "museums and herbaria", is that on purpose? Is ANIC in scope?

L73-74 The case for critical dependency between spatio-temporal representation of patterns of biodiversity being underpinned by collecting activities has not been made. You could reword to say that collections are foundational to that activity, but you should acknowledge that the bulk of that kind of work is (for better and worse) being done on the basis of observational data (~ 90% of datasets like GBIF). As I mentioned earlier, I think it would be better to focus on the kinds of inference that specimens can uniquely inform, and then those to which they also contribute as an important minority.

L85 and elsewhere. There are many references to your 'comprehensive' analysis of GBIF. I suggest you remove them and trust that your commendable work speaks for itself.

L253—255 If it is beyond the scope of the paper, then reduce this to ", probably for different reasons." at the end of preceding sentence?

Figure 4 - I think this diagram could be removed, or should be carefully revised. The caption perhaps stands alone but the diagram itself is problematic. What does "Impact FROM collection science" mean? AI tools emerge at the same point in time as the maximum taxonomic and curatorial capacity? If "demand for collection data" were increasing presumably it would be supplied, so the question is demand from whom? Global environmental issues is too generic. Microplastics?

L282 - Potential drivers of decline in collections data

I'd have thought that the explosion in autonomous sensor tech (camera trapping, audio samples) and citizen science data is a source of diminished perception of the value of specimen collection? Interacts with the ethics argument, in that the bar become higher to justify physical collection.

I invite the authors to break this right down to help build a conceptual model of how the processes that generate specimen collection have they changed over time, and how that may or may not result in a record in a database like GBIF, or perhaps to a generic source digital compendium that then gets contributed to GBIF. This model diagram could replace Fig 4. What forces accelerate collection activity versus diminish it - there might be some phases of colonial exploration, bio-prospecting, documentation of destruction at development, bio-regional conservation planning, some science for science sake sprinkled in....

L290 - surely developed-world labour costs should be mentioned here, and indicated as a possible reason why there appears to be relatively more collection in LMIC later in the analysed sequence. Also, at some point around the late 90s solo field work became untenable. Combining these it is not just decreased funding but massively higher cost per unit collection, flowing through to curation, etc.

L294 - This wouldn't apply to plant specimen collections though, and they were already declining?

L316 - To me the authors with this very paper should do their utmost to improve the public understanding of the unique value of collections. In the present version there is a sense of overreach because spatial and temporal representation of biodiversity trends is claimed as a value of these collections, whereas (again, for better or worse) those representations are mostly being done on the basis of occurrence data that resulted in no physical specimen being collected.

L321 - surely you can find references to support this paragraph?

L330—333 Here you get the expression just right it is the cumulative value of the collection as a comprehensive and authoritative resource that is eroding not the collection itself. Please trace expressions of this kind back through the article and correct other instances where you imply that the resource and not the value is eroding.

L337—343 Surely the bulk of scientific publication on these themes is being done with observational data (and again, for better and worse). The authors need to either address this head on with a critique of the bulk of that activity, making the case for the need to step up specimen collection, or substantially temper the tone of this section.

L346 fundamental?

L358—367 Might be a good place to discuss when specimens are required and when observational data might suffice?

L385—392 I'm afraid I can't quite grasp what is being suggested here.

L415 - the accession backlog / delay question is fundamental to your analysis and I believe it should be discussed in the main text.

L418—427 Lots of fairly generic assertions of strength and robustness in this para - consider revising. I think more directly discussion and probable dismissal of the possibility of a rival explanation of accession would be more convincing.

L429—443. I suggest that you rewrite this conclusion, after cleaning up the case for the unique value of collections in the main article. As much as physical collections are fundamental to our understanding of the science of biodiversity and evolutionary ecology, and as much as analyses based on observational data may sit unsteadily on that foundation, I would say you haven't demonstrated that it is imperative to continue to collect physical specimens at scale, and you seem to be implying that other forms of biodiversity occurrence data are not 'actionable'. They are, thanks to the legacy of the kind of datasets you analyse. Will they become less so as a result of reduced collection? That's the argument you are seeking.

L437-443. This para is over the top in my view. Biodiversity and ecosystem services are fundamental to life on earth. Natural history research collections are not, even if they represent the epistemological basis of our appreciation and management of both biodiversity and ecosystem services. I would remove this paragraph and start again once you have the rest tidied up.

(Remarks on code availability)

The code to reproduce the analyses seems similarly well documented, though I have not attempted to reproduce the code. I followed it through to understand how the occurrence downloads were filtered.

I presume the whole work-flow is not reproducible, since the X-axis labels on multi-panel Figure 2 are incorrect. That must have been edited after the fact in a image manipulation program.

Overall, the code and instructions are presented in way that would be instructive and encouraging to hypothetical colleagues looking to reproduce the work.

Reviewer #2

(Remarks to the Author)

Review Nature Comms
Forbes et al.

This is an important and timely perspective that I think raises many thought provoking points while also providing new

analyses and modeling that demonstrate that there is a decline in collection activity. I think this manuscript is a strong contribution. The advance encompassed in this manuscript is strong data analyses that evaluate collecting trends and the use of forecasting models to quantify these trends. This is a valuable contribution that goes beyond what has been done in previous work and provides new inference into temporal trends that have not been evaluated. For example, although Rohwer et al. 2022 also used GBIF data to assess collecting trends, they only focused on terrestrial vertebrates, while this paper also includes information on fish, arthropods, and plants, and in itself this is a useful contribution. The use of ARIMA models is novel. I found the ARIMA forecasting models particularly compelling and informative, but a few additional sentences describing the ARIMA forecasting would be helpful (currently Lines 149-162) to better orient the reader to the approach. The spatial context of the collecting trends is also noteworthy. The inclusion of the scripts and data are a strength of the manuscript.

I think the authors nicely outline the innovations in collections and metadata management that have helped spur new research and insight into a variety of issues of contemporary interest. The manuscript reviews some of the exciting innovations in curation digital specimens etc) and collections-driven research that has been used to infer new information addressing pressing global issues. The authors do a good job pointing out the utility of natural history collections to disciplines outside of taxonomy and evolution. This point has been made recently in other recent papers for example Nachman et al. 2023 (PlosBiology) which should be cited here as it similarly outlined why collections are important for myriad reasons. With regard to declines in collecting, some aspects of this topic have been addressed in other recent papers, in particular Rohwer et al. 2022 (PlosBiology) which is not cited here and should be. Rohwer et al. 2022 also includes efforts to quantify collecting trends (e.g. figure 1) using GBIF data. So, there are other recent papers that make similar points such as and these contributions should be cited in this manuscript.

The authors posit some reasons for the decline and I think expanding this to more fully encompass the "why" of the decline would be meaningful. For example, the authors could consider including a more explicit discussion on limitations that NHCs face e.g. funding for example -the reality is that most collections are run on small tight budgets and there is not much help for curatorial staff salaries from grant agencies beyond infrastructure-focused grant efforts. It is insufficient given the role that collections-based science could be driven if resources were available. Given more support, many NHCs would likely be very willing to engage in more innovative opportunities, but the day to day management and maintenance of the collection is often all that is feasible.

I would like to see solid recommendations for reversing the collections trend. What efforts and infrastructure are needed? The authors outline a vision for "collecting smarter" but this section just scratches the surface. Multi-institution efforts have led to innovations and adoption of emerging technologies already (e.g. iDigBio for example) so what do the authors recommend NHCs do additionally to "integrate and embrace" emerging technologies? The statement although valid is vague. Discussing in more detail how a specific infrastructure advance has spurred advances might be a way to get at this. For example, the origination of GBIF would perhaps be fitting since these aggregated data form the basis of the analyses presented here and this initiative helped spur digitization efforts.

(Remarks on code availability)

I reviewed the code files but given the large datasets used I did not run through the pipeline. The code appears to be well annotated.

Reviewer #3

(Remarks to the Author)

I was very excited to see this paper. In conversations with colleagues, we have discussed an anecdotal decline in deposition of new specimens into natural history collections and thus it is nice to see such a data-rich analysis of this issue. In general, I think this is an important and timely paper, but I would like to ask authors to address a few issues.

1. Impacts of specimen backlogs and data lags, particularly in Arthropoda. My experience with insect collections suggests that newly collected specimens are more likely to be prioritized for addition to digital databases by active researchers, while older pinned specimens are only included when part of specific digitization initiatives or when they are in a published paper (e.g. Figure S12). This leads to two potential issues for this paper – (1) it takes time (often years) for new specimens to be processed and added to digital repositories, causing GBIF records to potentially underestimate of specimens collected in most recent years. (2) GBIF records are likely to under-report the number of older specimens in collections. Chordata collections are typically smaller, and herbarium specimens are easier to digitize, so this issue will be most relevant for Arthropoda. While the authors do discuss this issue extensively in the supplemental material, I am skeptical of inferring trends from very recent records (see comment below).

2. Impacts of the pandemic on data collection and reporting. A major finding of this paper is a sharp decrease in GBIF specimen records post-2020. However, this coincides with the COVID-19 pandemic, which caused reduced collecting trips, and reductions in staff and other support of collections. These issues likely exacerbated backlogs (see above), and these effects may still be felt by collections managers today. In general, the impact of the pandemic felt underdiscussed throughout.

3. Considering the above, I think the main figures in the paper should be restricted to 1950-2019, with full dataset in the supplemental material. Similarly, I would like to see the forecasting model repeated using December 2019 as the final date to see if avoiding the pandemic changes any predictions for trends in collections. Alternatively, the authors may attempt to quantify the impact of backlogs by some compensatory mechanism. This could be achieved by comparing the number of specimens collected and archived in 2020 (or another recent year) to the number of specimens collected in 2020 but archived at a later date.

4. Changes in spatial trends in collecting. While this is a very large dataset, many collections do not contribute records to GBIF. I have some concern that the reported declines in collections in some regions of the world may be due to changes in North America/Europe collections that contribute greatly to GBIF decreasing collections in other countries, while developing countries increase their collections but are less likely to contribute records to GBIF. Is there any data on the geographic distribution of collections that are contributing to GBIF? This would make a nice supplemental figure showing the locations of these collections.

5. Figure 4 could use some polishing. The placement of the text was difficult to read in parts and some of the distributions of labels were unclear.

6. Additional citations are needed in the "potential drivers of declines in collection data" section (Lines 282-325) to reflect recent literature on these topics.

7. Line 385-392 – In this paragraph, you suggest that the responsibility for promoting the value of natural history collections should fall primarily to collection scientists. Given the reduction in collection scientist staff size and the growth of the value in collections in other areas, this should be the responsibility of the wider scientific community.

(Remarks on code availability)

The authors have a well-organized Zendo repository containing the code to clean and create figures.

Reviewer #4

(Remarks to the Author)

(Remarks on code availability)

Version 1:

Reviewer comments:

Reviewer #1

(Remarks to the Author)

I have reviewed the Authors responses to each of the reviews and the adjusted manuscript. I think their responses are thoughtful, detailed and diligent and it is clear to me that they've taken the opportunity to improve the clarity of key arguments and hone the text throughout.

In their rebuttal the authors are clear to thank all of the reviewers for their efforts that were instrumental in making this a much more compelling piece. There is no record of that gratitude in the Acknowledgments at present. I really hope that they rectify that.

(Remarks on code availability)

I did not return to review the code in this second phase.

Reviewer #3

(Remarks to the Author)

I thank the authors for their thoughtful revisions.

(Remarks on code availability)

Reviewer #4

(Remarks to the Author)

I co-reviewed this manuscript with one of the reviewers who provided the listed reports. This is part of the Nature

Communications initiative to facilitate training in peer review and to provide appropriate recognition for Early Career Researchers who co-review manuscripts.

(Remarks on code availability)

REVIEWER COMMENTS

Reviewer #1 (Remarks to the Author):

Forbes and colleagues are concerned that the rate of collections of biological specimens are declining across the world and across all taxonomic groups, just at a moment when new technologies in genomic analyses, imaging, and AI are enabling the extraction of data that is uniquely available from preserved specimens at scales (numbers of records) previously unthinkable. Their analysis of the records of preserved-specimens contained in the Global Biodiversity Information Facility (GBIF) database suggests just that. The global peak in specimen collection considered as unique species records per collection-year for vertebrates (1960s), plants (1980s) and arthropods (~2010) are all past, and the spatial spread of those observations is also declining, although the trace of a concerted effort to prioritise undersampled areas results in a resurgence and delayed decline when the records are considered in spatial terms. That latter is an interesting addition to a simple consideration of the fate of collections from a collection-year only perspective.

Thank you for your valuable feedback. We have responded to each of your comments below.

The patterns they find globally seem to agree with those previously published with GBIF vertebrate data across by Rohwer et al (2022, <https://doi.org/10.1371/journal.pbio.3001613>), Malaney and Cook 2018 for mammal collections in the US (2018, <https://doi.org/10.1093/jmammal/gyy082>), and Prather et al for herbaria records in US (2004, <https://www.jstor.org/stable/25063934>)(<https://research.ebsco.com/linkprocessor/v2-external?opid=xppotz&recordId=tqryfze4db&url=https%3A//www.jstor.org/stable/25063934>)). Forbes et al's work is clearly more comprehensive in geographic, temporal and taxonomic breadth than the three articles I mention, and the use of additional response variable formulations including geographic coverage per broad taxonomic group in this paper make it a more interesting exploration of the data trends. Nonetheless, this work would be improved by acknowledging existing literature that is surely not limited to the three I found above with a quick search. The GBIF platform itself provides some simple graphical analysis of its holdings that readers could be referred to (<https://www.gbif.org/analytics/global>).

Thank you for highlighting this relevant literature. We have included references to these recommended articles in our Introduction in the following paragraph at L100:

“Given the critical dependency on spatiotemporal representation of global patterns in biodiversity underpinned by collecting activities, it is important to explore trends in how herbaria and museums are adding specimens, representing diverse species, geographical locations and environments, and how these collecting trends are changing over time. Previous studies have identified patterns of decline in specimen collection across specific taxonomic groups and regions. For example, Prather et al.

documented a significant decline in plant collecting in the United States¹⁴, while Malaney and Cook highlighted reduced availability of mammalian specimens as an issue of particular concern during the present era of rapid environmental change¹⁵. More recently, Rohwer et al. demonstrated declining growth in vertebrate collections globally using GBIF data¹⁶. Building on these important contributions, we conducted a comprehensive global analysis across major taxonomic groups.”

The authors start by referencing all the museums and herbaria of the world, and then switch to GBIF, almost as though it was effectively the same thing (accession aside). Do all institutions submit their records to GBIF? Does ANIC? Maybe I should know the answer to the former but I don't, so I think the authors should clarify for the sake of readers.

Thank you for your feedback. We have added the following clarification at L583:

“GBIF data represent only a subset of global natural history collections, as participation is voluntary and most institutions have incomplete digital coverage. Our study specifically filtered for preserved specimens (approximately 10% of GBIF's total holdings) to focus on specimen collection trends rather than broader biodiversity observation patterns.”

For the sake of clarity and avoiding repetition, below we have grouped together Reviewer 1's comments regarding specimens vs. observational data and other ecological data sources.

I suspect that the average reader is likely to mis-interpret the findings presented here, failing to pick up on the authors' subtle framing and deft avoidance of mentioning observation data, including that from citizen scientists, which account for about 90% of GBIF's records. I think they should insert a sentence that explains what GBIF is and where preserved specimens fit in amongst the types of records held there, in the methods if not earlier.

I think the discussion and main claims are problematic at present because the authors don't mention observational data, including but not limited to structured citizen science data, until late in the Discussion (L398). Those data are (for better and worse) far more numerous, and consequently more commonly employed in main of the domains that the authors claim are at risk from declining collections, e.g. from L339-340, “estimate ongoing climate and land-use change impacts, model species distribution trends, and predict future ecological ... impacts [all for estimating spatial and temporal trends to support evidence based decisions on conservation, ecosystem services and crisis management]. While the authors have cited a couple of published examples where specimen data were used for those purposes, one could find thousands where those objectives were being pursued using occurrence data that didn't relate to a collected specimen. The authors need to either address this head on with a critique of the bulk of that activity, making the case for the need to step up specimen collection, or they need to substantially reduce the scope of the claims to those for which physical collections are fundamental.

The future directions section is strong, and could easily include consideration

of how physical and observational or remotely captured / sensed data could be combined, or how specimen collection might respond to the far more numerous observational datasets.

L73-74 The case for critical dependency between spatio-temporal representation of patterns of biodiversity being underpinned by collecting activities has not been made. You could reword to say that collections are foundational to that activity, but you should acknowledge that the bulk of that kind of work is (for better and worse) being done on the basis of observational data (~ 90% of datasets like GBIF). As I mentioned earlier, I think it would be better to focus on the kinds of inference that specimens can uniquely inform, and then those to which they also contribute as an important minority.

L316 - To me the authors with this very paper should do their utmost to improve the public understanding of the unique value of collections. In the present version there is a sense of overreach because spatial and temporal representation of biodiversity trends is claimed as a value of these collections, whereas (again, for better or worse) those representations are mostly being done on the basis of occurrence data that resulted in no physical specimen being collected.

L358—367 Might be a good place to discuss when specimens are required and when observational data might suffice?

Thank you for your valuable feedback. We agree that our original manuscript did not sufficiently articulate this critical distinction, which risked misleading readers about the scope and implications of our findings. Your observations highlighted important opportunities to strengthen our manuscript.

In response, we have made substantial revisions throughout the paper to clearly delineate the unique value of specimen collections while acknowledging their complementary relationship with observational data streams. We have added text in the Introduction (L81) that explicitly defines our focus on specimen-based records and articulates their irreplaceable scientific value for taxonomy, ground-truthing of other data streams, and novel data extraction. We have also expanded our discussion of how specimen collections critically underpin other ecological data sources (L431), clarified the impacts of decreased specimen coverage on broader ecological inferences (L457), and expanded our Future Directions section (L479) to address the integration of specimen data within the broader biodiversity information ecosystem. We believe these changes have strengthened the manuscript considerably by providing a more nuanced and accurate representation of the unique role of specimen-based data in biodiversity science, and why the trends we observe should be of concern.

In the Introduction at L81, we have added this section which emphasises the critical dependence of observational data (especially citizen scientist observations) on specimen collections, which is why we focus on the latter:

“While ecological data from field observations, citizen science and remote sensing continue to grow, specimen-based records remain crucial as they enable accurate

species identification, verification and quality assurance for other data streams, and they possess unique historical extent and temporal continuity. In addition to these essential functions of ground-truthing and verification, as Nachman et al. highlight specimens are essential for discovering new species, tracking environmental degradation, studying morphology and physiology, investigating gene expression and epigenetic modifications associated with environmental adaptation, and extracting novel information as analytical technologies continue to advance¹³. This capacity for repeated examination and data extraction makes specimen collections an irreplaceable scientific resource with unique value beyond what observational data alone can provide.”

At L431, in the ‘Drivers of Decline’ section, we have expanded the following paragraph:

“e) Funder and public perceptions: There continue to be persistent gaps in public awareness and funding body understanding of the role of natural history collections or how sampling, identification, and preservation of physical specimens critically underpins the ability of collections to contribute to addressing environmental problems³¹. Importantly, collections-based work by formally trained taxonomists provides the essential knowledge base for other data sources such as long-term ecological sites, citizen science initiatives, and indeed any aspect of biodiversity science³².”

At L457, in the ‘Potential Impacts On Collection Science’ section, we have added the following sentence:

“Decreased specimen data coverage leads to lower accuracy and higher uncertainty in statistical inferences from collections data, and also undermines the utility of other ecological data streams which are dependent on accurate species identification.”

At L479, in the ‘Future Directions’ section, we have added the following discussion of the role of specimen-based data in the context of a broader ecology data ecosystem consisting of observational and other data streams:

“As a scientific discipline, we need to take action to address this problem. To fully realise the future scientific value and impact of natural history collections we must first look critically at their role going forward, not in isolation but as part of a dynamically evolving biodiversity information ecosystem. This ecosystem is being rapidly populated by a diverse range of emerging data streams including citizen science observations, satellite and remote sensing information, eco-acoustics and environmental DNA. Each of these delivers valuable data but is associated with its own biases and errors. Focusing on the key attributes of specimen-based collections (spatial coverage, temporal depth and taxonomic certainty and breadth) provides the basis for assessment of what information specimens can uniquely deliver as well as their complementarity to these other data sources. This will be achieved most usefully when framed by the specific data needs of emerging application domains. This involves considering what types of data are most valuable for specific research questions, how spatiotemporal scale relates to different research domains (e.g., species discovery, eco-evolutionary processes, climate and land-use change impacts), and which analytical tools can maximise the impact-to-effort ratio in

research prioritisation. Importantly, these considerations apply not only to new specimen collection but also to increasing and optimising the databasing and digital mobilisation of existing specimen backlogs, particularly for hyper-diverse groups like Arthropoda, and to choosing which data layers (e.g. genomic, phenotypic, environmental) are most valuable to add.”

GBIF is used as the source data for this study presumably because of its global reach and the enormous effort already invested by contributors and curators in standardisation. The downside as they acknowledge is that they don't know about accession dates (how on earth is that not a mandatory field?), so no stock-and-flow kind of model can be built that would allow estimation of bottlenecks applicable to different taxonomic groups, and in different country or regional settings. I think the authors should clarify what their assumptions are that supply process. They warn that longer delays between collection and accession are expected for arthropods but a clear statement about what they assume is the case for each taxonomic group is probably warranted so that readers can keep that in mind when considering the results. For Arthropods, since they are attached to the Australian National Insect Collection, perhaps they could use it as an example and describe the proportion of records ANIC holds that are in GBIF? I think the authors are correct in their assumption about declining collection, but without accession dates the dataset is something of a black box within which declining collection and accession remain rival explanations, no? I think the authors should do what they can to enlighten readers about what the dataset contains and what it is missing - if there is no literature or way to do that then that is an interesting additional finding. [Footnote, my internet search on the topic just found a forum question on GBIF site that could well be from the lead author “of2”- so they have clearly tried to untangle accession and collection - now they just need to communicate the finding in the main paper as it is something that all readers should take away without having to dig in the supplements.]

Thank you for your valuable comments. We agree that it is surprising that accession date is not recorded in GBIF, and this limited our ability to develop a ‘stock-and-flow’ style model, or to disentangle the separate contributions of decreased collecting, and time lags between specimen collection, local databasing, and accession to GBIF. Indeed that thread on the GBIF forum was part of our efforts to source such data, alongside discussions directly with GBIF staff.

We have addressed these issues around ARIMA modelling being unable to disentangle these potential contributing effects and included details in the main body of the manuscript instead of the supplements, as suggested. We have added the paragraph below from L587:

“A significant data limitation and analytical challenge is the absence of accession dates in GBIF records. Without these dates, we cannot fully disentangle three distinct processes: (1) actual declines in specimen collecting activity, (2) time lags between collection and local databasing, and (3) delays between local databasing and GBIF contribution. To address this data limitation, we implemented forecasting analyses using historical GBIF database snapshots. See Sections 5 and 6 of the Supplementary Information for a detailed discussion of these data challenges.”

The authors combine raw data plots with autoregressive integrated moving average (ARIMA) models to project the trend forward in time, which seems appropriate. If accession dates were available then perhaps a hierarchical model separating collection and accession could have been considered. The analysis performed is well documented in the article and at first blush the code required to reproduce the analyses seems similarly well documented. It is presented in way that would be instructive to hypothetical colleagues. Considerable effort has gone into harmonising data and removing potentially artifactual elements.

The plots of temporal trends in collections-by-year in Figure 1 are simple but compelling. I trust they are backed up by the forecasting analysis, handsomely presented in Fig 2 (though the X-axis must be mis-labelled as they end at 2020?) that also suggest ongoing decline, amidst considerable uncertainty. I think the authors should chose either the 5 year or the 10 year forecasts, placing the other in Suppl if desired.

Thank you for your positive feedback. The forecasting results in Figure 2 cover collection years up to 2019. We also note that in response to a comment from Reviewer 3, we have now limited the range of collection years for both Figures 1 and 2 to end at 2019, to avoid over-interpreting trends in recent years impacted by the COVID-19 pandemic, with only a short period of time elapsed since data collection.

We feel that including both the 5-year (2028) and 10-year (2033) forecast horizons adds value for the sake of comparison. For instance, the reader can compare side by side and observe that the uncertainty is greater for recent collecting years, with higher uncertainty at the 10-year forecast. While the decline trend may flatten somewhat at 10-year relative to 5-year forecasts, overall, the declines are still forecast to remain with ongoing accessions to GBIF.

L21 Consider 'Our analysis of over X records from GBIF spanning more than two centuries suggests (significant) declines in the **rate of collection of specimen data over recent decades.'

Thank you for the suggested change. We have amended the sentence at L21: "Here we show substantial declines in the rates of collection of specimen data over recent decades, from analysis of over 150 million records from the Global Biodiversity Information Facility (GBIF) spanning more than two centuries."

L25-26 You don't show "reduced availability" right, data are not being removed from GBIF and accession continues, increasing accessibility (L37). I have understood your finding to be that the source (collecting activity) has been dramatically constricted and that signal is appreciable despite an unknown but generally constant effect of variable accession. Or do you think that commitment to accession is declining, in which case the article would need to be substantially modified to indicate that in the end you still have two important rival explanations, both of which are concerning but have different implications.

Thank you for your comment. To clarify the phrasing, we have amended the sentence at L25 as follows:

“Overall, these findings suggest that the potential applications for natural history collections as global research infrastructure is eroding due to decreased collecting of specimen data across species, locations, and time.”

L29 Consider using specimens for natural history, now that you have linked the two ideas in the first sentence.

Thank you for the suggestion. In this case, we prefer to keep the current institutional-level focus for this sentence.

L31 Here instead of natural history you are using “museums and herbaria”, is that on purpose? Is ANIC in scope?

Thank you. We have amended these sentences for clarity:

“The world’s natural history collections contain between 2 and 4 billion preserved specimens of plants, animals and fungi dating back over 400 years^{1,2}. Together these collections, represent the most comprehensive taxonomic and spatial record of the Earth’s biodiversity and its change through time, which is increasingly driven by accelerating ecological crises.”

L85 and elsewhere. There are many references to your ‘comprehensive’ analysis of GBIF. I suggest you remove them and trust that your commendable work speaks for itself.

As suggested, we have removed reference to our analysis as ‘comprehensive’ here and in several other locations in the manuscript.

L253—255 If it is beyond the scope of the paper, then reduce this to “, probably for different reasons.” at the end of preceding sentence?

We have amended these sentences at L352:

“For instance, in Europe, Chordata collecting peaked between 1965-1985, while Plantae peaked around 2010. Within Chordata, collecting in both Africa and North America peaked in the 1960s and showed subsequent sharp declines, albeit probably for different reasons.”

Figure 4 - I think this diagram could be removed, or should be carefully revised. The caption perhaps stands alone but the diagram itself is problematic. What does “Impact FROM collection science” mean? AI tools emerge at the same point in time as the maximum taxonomic and curatorial capacity? If “demand for collection data” were increasing presumably it would be supplied, so the question is demand from whom? Global environmental issues is too generic. Microplastics?

Thank you for your feedback. As multiple reviewers raised concerns with Figure 4, we have removed it from the manuscript.

L282 - Potential drivers of decline in collections data. I'd have thought that the explosion in autonomous sensor tech (camera trapping, audio samples) and citizen science data is a source of diminished perception of the value of specimen collection? Interacts with the ethics argument, in that the bar become higher to justify physical collection. I invite the authors to break this right down to help build a conceptual model of how the processes that generate specimen collection have they changed over time, and how that may or may not result in a record in a database like GBIF, or perhaps to a generic source digital compendium that then gets contributed to GBIF. This model diagram could replace Fig 4. What forces accelerate collection activity versus diminish it - there might be some phases of colonial exploration, bio-prospecting, documentation of destruction at development, bio-regional conservation planning, some science for science sake sprinkled in....

Thank you for this valuable feedback. We agree that the explosion in autonomous sensor technology and citizen science data has significantly influenced perceptions around specimen collection, and have expanded point (b) in our 'Drivers of Decline' section to address this directly at L399:

“b) Regulatory and ethical considerations: These include safety/risk assessments (e.g., biosecurity and conservation concerns), which increasingly drive approval processes for collection permits. The rapid growth of alternative data streams (e.g., autonomous sensors, audio recordings, citizen science observations) likely contributes to views that the continued collection of physical specimens needs much stronger scientific justification than would have been the case in the past. This is particularly the case for vertebrates, where public sensitivity to collecting activities is high. Emerging national and international initiatives aimed at recognising the rights of traditional owners (e.g., the Nagoya Protocol, the Global Indigenous Data Alliance, CARE principles) add further complexities, requiring thoughtful navigation²⁵⁻²⁷.”

Your suggestion for a conceptual diagram mapping the historical forces that have influenced specimen collection (colonial exploration, bio-prospecting, conservation planning, etc.) and their pathways to digital repositories is compelling. While we appreciate the value such a model would add, developing a robust conceptual framework would require quantifying the relative importance and interactions of these complex processes across different time periods, regions, and taxonomic groups—an analysis that extends beyond our current scope.

We do acknowledge the importance of understanding these drivers in our introduction to the 'Potential Drivers' section (L375):

“There are likely multiple causal factors associated with the observed decline in collecting activity (e.g., geopolitical, socioeconomic, support for scientific and digital infrastructure, resources for growing and training capability, shifting public perceptions etc.), the relative importance of which will vary by country and institution. While analysis of how various drivers may interact at institutional, local, national and global scales to drive observed collecting patterns is beyond the scope of this study, we highlight some key considerations here”

We believe this conceptual framework would be valuable as a dedicated follow-up study that could more thoroughly investigate the historical and contemporary forces shaping specimen collection practices.

L290 - surely developed-world labour costs should be mentioned here, and indicated as a possible reason why there appears to be relatively more collection in LMIC later in the analysed sequence. Also, at some point around the late 90s solo field work became untenable. Combining these it is not just decreased funding but massively higher cost per unit collection, flowing through to curation, etc.

Thank you for raising this important point. We have amended this paragraph to reflect this, as follows (L392):

“a) Resource constraints: Limited funding sources for collections, coupled with rising labour and infrastructure costs, and stricter fieldwork safety requirements have dramatically increased per-specimen costs and reduced institutional capacity for collecting, processing, and housing specimens^{22–24}. For many natural history collections, ever-declining budgets means that basic maintenance and collection management may be all that is feasible, despite growing demands for specimen-based data.”

L294 - This wouldn't apply to plant specimen collections though, and they were already declining?

Thank you for your comment. We have adjusted this section with the following sentence at L404:

“This is particularly the case for vertebrates, where public sensitivity to collecting activities is high, though is not as relevant as a driver to consider for plants.”

L321 - surely you can find references to support this paragraph?

Thank you for your feedback. In response to this comment and to feedback from other reviewers, we have added citations throughout this section to support our claims in each of these paragraphs (a) to (f) in the ‘Potential Drivers Of Decline’, including the following paper specifically regarding approaching taxonomic saturation for certain regions:

33. Moura, M. R. & Jetz, W. Shortfalls and opportunities in terrestrial vertebrate species discovery. *Nat Ecol Evol* 5, 631–639 (2021).

L330—333 Here you get the expression just right it is the cumulative value of the collection as a comprehensive and authoritative resource that is eroding not the collection itself. Please trace expressions of this kind back through the article and correct other instances where you imply that the resource and not the value is eroding.

Thank you for the positive feedback. Throughout the manuscript, we have been careful to articulate that the rate of collecting/growth in specimen data is decreasing, resulting in an erosion of the value and utility of this resource. We are not making the claim that the total volume of specimen data is decreasing. For example, at L21:

“Here we show substantial declines in the rate of collection of specimen data over recent decades...”

L26:

“Overall, these findings suggest that the value of natural history collections as global research infrastructure is eroding due to decreased collecting of specimen data across species, locations, and time.”

L474:

“While increasingly, collections have new data science tools and technical platforms at their disposal, the trend of decreasing growth in collection data inhibits their capacity to leverage these tools to their full potential...”

L337—343 Surely the bulk of scientific publication on these themes is being done with observational data (and again, for better and worse). The authors need to either address this head on with a critique of the bulk of that activity, making the case for the need to step up specimen collection, or substantially temper the tone of this section.

Thank you for your valuable feedback. Connecting with our other responses above and revisions throughout the paper regarding the benefits and characteristics of specimens vs. observational data, we have modified this paragraph at L459:

“Decreased specimen data coverage leads to lower accuracy and higher uncertainty in statistical inferences from collections data, and also undermines the utility of other ecological data streams which are dependent on accurate species identification. These trends are compromising our ability to estimate ongoing climate and land-use change impacts, model species distribution trends, and predict future ecological and evolutionary impacts. This data gap further increases the risk that we will be unable to adequately manage or protect the ecosystems that represent our greatest natural resource, providing critical life support systems for humanity and Earth’s biosphere.”

L346 fundamental?

Thank you. We have amended the sentence at L469:

"Importantly, for natural history collections to mobilise, enhance and add value to the specimen data they hold, it is critical that they also deploy and integrate emerging technologies."

L385—392 I'm afraid I can't quite grasp what is being suggested here.

In response to this comment and feedback from other reviewers, we have revised and expanded this paragraph (now at L530) to add clarity:

“The scientific community as a whole needs to collaboratively advocate for the value of digitised and mobilised collections data in tackling global environmental problems. While collection scientists bring crucial expertise, the growing applications of specimen data across diverse disciplines necessitate broader engagement from researchers in ecology, genomics, climate science, and other fields. Building awareness of the tremendous value inherent in natural history collections, both in the

public (e.g., via citizen science initiatives) and in funding bodies is crucial for securing resources needed to grow collections and address structural and capability constraints. Similarly, structured exploration of the data needs of relevant research disciplines and applications will help to shape what collections data should most usefully look like in the future.”

L415 - the accession backlog / delay question is fundamental to your analysis and I believe it should be discussed in the main text.

L418—427 Lots of fairly generic assertions of strength and robustness in this para - consider revising. I think more directly discussion and probable dismissal of the possibility of a rival explanation of accession would be more convincing.

Thank you for your feedback. Drawing material from the supplementary information, we have added the paragraph below from L587:

“A significant data limitation and analytical challenge is the absence of accession dates in GBIF records. Without these dates, we cannot fully disentangle three distinct processes: (1) actual declines in specimen collecting activity, (2) time lags between collection and local databasing, and (3) delays between local databasing and GBIF contribution. To address this data limitation, we implemented forecasting analyses using historical GBIF database snapshots. See Sections 5 and 6 of the Supplementary Information for a detailed discussion of these data challenges.”

Following on from this paragraph, we have also revised the following paragraph as suggested (previously from L418-427), now at L545:

“Despite these caveats, the patterns we have identified are consistent across major taxonomic groups and geographic regions. Our study, based on over 150 million datapoints from numerous collections worldwide, provides a robust foundation for our conclusions. While acknowledging the inherent limitations around data quality and comprehensive cleaning for aggregated biodiversity data³⁶, the overall patterns in our study remain substantial. The temporal consistency and taxonomic breadth of our findings, coupled with ARIMA forecasting results, suggests that the observed declines in specimen collecting represent a real phenomenon rather than an artifact of data management processes. These findings underscore the urgent need for continued investment in and strategic planning for natural history collections to ensure their vital role in addressing global environmental challenges.”

L429—443. I suggest that you rewrite this conclusion, after cleaning up the case for the unique value of collections in the main article. As much as physical collections are fundamental to our understanding of the science of biodiversity and evolutionary ecology, and as much as analyses based on observational data may sit unsteadily on that foundation, I would say you haven't demonstrated that it is imperative to continue to collect physical specimens at scale, and you seem to be implying that other forms of biodiversity occurrence data are not 'actionable'. They are, thanks to the legacy of the kind of datasets you analyse. Will they become less so as a result of reduced collection? That's the argument you are seeking.

L437-443. This para is over the top in my view. Biodiversity and ecosystem services are fundamental to life on earth. Natural history research collections are not, even if they represent the epistemological basis of our appreciation and management of both biodiversity and ecosystem services. I would remove this paragraph and start again once you have the rest tidied up.

Thank you for your feedback. We have revised the Conclusion as follows, to explicitly incorporate discussion of the unique value of specimen-based data in the context of a broader ecosystem of ecological data streams including observational data:

“In an era marked by rapid climate change and biodiversity loss, it is imperative that we continue to invest in the unique value of natural history collections data. We argue that such resources not only directly contribute to addressing the increasingly urgent need for actionable ecological information but also underpin other critical biodiversity and environmental data sources (e.g., citizen observations, remote sensing, genetic information) that depend on accurate species identification. Thus, the decline in collections data is a worrying trend that requires strategic approaches to how we resource and target ongoing collection efforts, to ensure that we can effectively meet the accelerating environmental challenges of our time.

To be explicit, we are not arguing for a return to the same levels of collecting activities that occurred in the past, but rather that we need to collect ‘smarter’, especially given ongoing resource constraints. Modern collecting practices need to reflect the expanding scope of applications for specimen data beyond traditional taxonomy, and embrace the diversity of information types that can now be extracted from each specimen through advances in digitisation, genomics and AI-powered trait extraction. In practical terms, this might mean objectively evaluating whether additional physical specimens are required for a particular research or application objective, or whether existing data might suffice. This determination is becoming increasingly important given the rapidly growing body of information from other data streams including citizen science and field observations, that now make up approximately 90% of observations in GBIF, alongside eDNA and remote sensing data. If new collecting activities are indicated, then sampling should aim to maximise the value of the additional information gained. This can be achieved in part, via the development and deployment of tools that aid in effective integration across data layers, and application of data-driven sampling design frameworks in the context of this broader data ecosystem.

We are only starting to fully appreciate the value of the irreplaceable data held in natural history research collections at a time when their unique spatiotemporal coverage is under threat. It is crucial that we work together to strengthen, protect and invest in specimen-based data collection for the sake of science, humanity and the planet. It is worth noting that while to some extent, we can recapture spatial information about species occurrences, we cannot recapture time. And time is something that our natural systems do not have.”

Reviewer #1 (Remarks on code availability):

The code to reproduce the analyses seems similarly well documented, though I have not attempted to reproduce the code. I followed it through to understand how the occurrence downloads were filtered.

I presume the whole work-flow is not reproducible, since the X-axis labels on multi-panel Figure 2 are incorrect. That must have been edited after the fact in a image manipulation program.

Overall, the code and instructions are presented in way that would be instructive and encouraging to hypothetical colleagues looking to reproduce the work.

Thank you for your positive comments. The workflow is fully reproducible, and all figures are generated directly from the R code without any further modification. The x-axis for Figure 2 covers the intended range up to 2019, as we described in response to your comment above.

Reviewer #2 (Remarks to the Author):

**Review Nature Comms
Forbes et al.**

This is an important and timely perspective that I think raises many thought provoking points while also providing new analyses and modeling that demonstrate that there is a decline in collection activity. I think this manuscript is a strong contribution. The advance encompassed in this manuscript is strong data analyses that evaluates collecting trends and the use of forecasting models to quantify these trends. This is a valuable contribution that goes beyond what has been done in previous work and provides new inference into temporal trends that have not been evaluated. For example although Rohwer et al. 2022 also used GBIF data to assess collecting trends, they only focused on terrestrial vertebrates, while this paper also includes information on fish, arthropods, and plants, and in itself this is a useful contribution.

Thank you for your valuable feedback. We have responded to each of your comments in detail below.

The use of ARIMA models is novel. I found the ARIMA forecasting models particularly compelling and informative, but a few additional sentences describing the ARIMA forecasting would be helpful (currently Lines 149-162) to better orient the reader to the approach. The spatial context of the collecting trends is also noteworthy. The inclusion of the scripts and data are a strength of the manuscript.

Thank you for your suggestion. To support reader understanding, we have added the following explanatory sentences from L203:

“ARIMA models are particularly well-suited for this analysis as they account for temporal dependencies in time series data while allowing for non-stationary trends²⁰. In the absence of accession dates for GBIF records, these models provide a robust alternative approach to understanding the timeline between specimen collection and their appearance in global databases. Each ARIMA model captures the historical pattern of how records for a specific collection year have accumulated in GBIF over time, driven by processes including new institutions sharing their collections to GBIF, and the overall timeline from new specimen collection, local taxonomic identification, and institutional databasing, through to eventual GBIF accession.”

I think the authors nicely outline the innovations in collections and metadata management that have helped spur new research and insight into a variety of issues of contemporary interest. The manuscript reviews some of the exciting innovations in curation digital specimens etc) and collections-driven research that has been used to infer new information addressing pressing global issues. The authors do a good job pointing out the utility of natural history collections to disciplines outside of taxonomy and evolution. This point has been made recently in other recent papers for example Nachman et al. 2023 (PlosBiology) which should be cited here as it similarly outlined why collections are important for myriad reasons. With regard to declines in

collecting, some aspects of this topic have been addressed in other recent papers, in particular Rohwer et al. 2022 (PlosBiology) which is not cited here and should be. Rohwer et al. 2022 also includes efforts to quantify collecting trends (e.g. figure 1) using GBIF data. So, there are other recent papers that make similar points such as and these contributions should be cited in this manuscript.

Thank you for your suggestions. We have included reference to Rohwer et al. 2022 and other similar papers in the following paragraph from L103:

“Previous studies have identified patterns of decline in specimen collection across specific taxonomic groups and regions. For example, Prather et al. documented a significant decline in plant collecting in the United States¹⁴, while Malaney and Cook highlighted reduced availability of mammalian specimens as an issue of particular concern during the present era of rapid environmental change¹⁵. More recently, Rohwer et al. demonstrated declining growth in vertebrate collections globally using GBIF data¹⁶. Building on these important contributions, we conducted a comprehensive global analysis across major taxonomic groups.

We have also included reference to Nachman et al. 2023 in the context of discussing the unique value of specimen-based data at L81:

“While ecological data from field observations, citizen science and remote sensing continue to grow, specimen-based records remain crucial as they enable accurate species identification, verification and quality assurance for other data streams, and they possess unique historical extent and temporal continuity. In addition to these essential functions of ground-truthing and verification, as Nachman et al. highlight specimens are essential for discovering new species, tracking environmental degradation, studying morphology and physiology, investigating gene expression and epigenetic modifications associated with environmental adaptation, and extracting novel information as analytical technologies continue to advance¹³. This capacity for repeated examination and data extraction makes specimen collections an irreplaceable scientific resource with unique value beyond what observational data alone can provide.”

The authors posit some reasons for the decline and I think expanding this to more fully encompass the "why" of the decline would be meaningful, For example The authors could consider including a more explicit discussion on limitations that NHCs face e.g. funding for example -the reality is that most collections are run on small tight budgets and there is not much help for curatorial staff salaries from grant agencies beyond infrastructure-focused grant efforts. It is insufficient given the role that collections-based science could be driven if resources were available. Given more support, many NHCs would likely be very willing to engage in more innovative opportunities, but the day to day management and maintenance of the collection is often all that is feasible.

Thank you for your feedback. In response to your suggestion and to feedback from other reviewers, we have expanded the ‘Drivers of Decline’ section with additional detail and citations. Specifically regarding your comments around resourcing and

budgets, while a thorough treatment of this topic is beyond the scope of this paper, we have expanded the following paragraph to incorporate this point more explicitly at L392:

“a) Resource constraints: Limited funding sources for collections, coupled with rising labour and infrastructure costs, and stricter fieldwork safety requirements have dramatically increased per-specimen costs and reduced institutional capacity for collecting, processing, and housing specimens^{22–24}. For many natural history collections, ever-declining budgets means that basic maintenance and collection management may be all that is feasible, despite growing demands for specimen-based data.”

I would like to see solid recommendations for reversing the collections trend. What efforts and infrastructure are needed? The authors outline a vision for “collecting smarter” but this section just scratches the surface. Multi-institution efforts have led to innovations and adoption of emerging technologies already (e.g. iDigBio for example) so what do the authors recommend NHCs do additionally to “integrate and embrace” emerging technologies? The statement although valid is vague. Discussing in more detail how a specific infrastructure advance has spurred advances might be a way to get at this. For example the origination of GBIF would perhaps be fitting since these aggregated data form the basis of the analyses presented here and this initiative helped spur digitization efforts.

Thank you for your valuable comments. We have expanded and added detail to the ‘Future Directions’ section to highlight possible paths forward in light of the observed decreasing trends in collecting, including at L481, L530, L547 and also in the conclusion from L567:

L481

“As a scientific discipline, we need to take action to address this problem. To fully realise the future scientific value and impact of natural history collections we must first look critically at their role going forward, not in isolation but as part of a dynamically evolving biodiversity information ecosystem. This ecosystem is being rapidly populated by a diverse range of emerging data streams including citizen science observations, satellite and remote sensing information, eco-acoustics and environmental DNA. Each of these delivers valuable data but is associated with its own biases and errors. Focusing on the key attributes of specimen-based collections (spatial coverage, temporal depth and taxonomic certainty and breadth) provides the basis for assessment of what information specimens can uniquely deliver as well as their complementarity to these other data sources. This will be achieved most usefully when framed by the specific data needs of emerging application domains.”

L530

“The scientific community as a whole needs to collaboratively advocate for the value of digitised and mobilised collections data in tackling global environmental problems. While collection scientists bring crucial expertise, the growing applications of specimen data across diverse disciplines necessitate broader engagement from researchers in ecology, genomics, climate science, and other fields. Building awareness of the tremendous value inherent in natural history collections, both in the

public (e.g., via citizen science initiatives) and in funding bodies is crucial for securing resources needed to grow collections and address structural and capability constraints. Similarly, structured exploration of the data needs of relevant research disciplines and applications will help to shape what collections data should most usefully look like in the future.”

L547

“Developing and extending data infrastructure is crucial for harmonising collections-based data with other ecological and environmental data sources, responding to declining collecting trends by maximising the value of each collected specimen. Existing examples of successful implementation of novel digital infrastructure include the data architecture standards developed by international platforms such as GBIF, DiSSCo and iDigBio³⁵. These technological innovations have spurred advances in collections science by enabling implementation of the digital extended specimen concept, transforming static records into dynamic, interconnected knowledge systems that integrate traditional specimen data with derived information like genomic sequences and phenotypic traits, alongside other ecological data layers.”

L624

“To be explicit, we are not arguing for a return to the same levels of collecting activities that occurred in the past, but rather that we need to collect ‘smarter’, especially given ongoing resource constraints. Modern collecting practices need to reflect the expanding scope of applications for specimen data beyond traditional taxonomy, and embrace the diversity of information types that can now be extracted from each specimen through advances in digitisation, genomics and AI-powered trait extraction. In practical terms, this might mean objectively evaluating whether additional physical specimens are required for a particular research or application objective, or whether existing data might suffice. This determination is becoming increasingly important given the rapidly growing body of information from other data streams including citizen science and field observations, that now make up approximately 90% of observations in GBIF, alongside eDNA and remote sensing data. If new collecting activities are indicated, then sampling should aim to maximise the value of the additional information gained. This can be achieved in part, via the development and deployment of tools that aid in effective integration across data layers, and application of data-driven sampling design frameworks in the context of this broader data ecosystem.”

Reviewer #2 (Remarks on code availability):

I reviewed the code files but given the large datasets used I did not run through the pipeline. The code appears to be well annotated.

Thank you for your time and feedback. We believe that our manuscript is much improved thanks to your thorough and helpful comments.

Reviewer #3 (Remarks to the Author):

I was very excited to see this paper. In conversations with colleagues, we have discussed an anecdotal decline in deposition of new specimens into natural history collections and thus it is nice to see such a data-rich analysis of this issue. In general, I think this is an important and timely paper, but I would like to authors to address a few issues.

Thank you for your valuable feedback. We have responded to each of your comments in detail below.

1. Impacts of specimen backlogs and data lags, particularly in Arthropoda. My experience with insect collections suggests that newly collected specimens are more likely to be prioritized for addition to digital databases by active researchers, while older pinned specimens are only included when part of specific digitization initiatives or when they are in a published paper (e.g. Figure S12). This leads to two potential issues for this paper – (1) it takes time (often years) for new specimens to be processed and added to digital repositories, causing GBIF records to potentially underestimate of specimens collected in most recent years. (2) GBIF records are likely to under-report the number of older specimens in collections. Chordata collections are typically smaller, and herbarium specimens are easier to digitize, so this issue will be most relevant for Arthropoda. While the authors do discuss this issue extensively in the supplemental material, I am skeptical of inferring trends from very recent records (see comment below).

Thank you for your comment. We agree that understanding time lags from collection to databasing and accession would aid understanding of these trends, and we were surprised to find that this information is not available in GBIF. We also agree with your point around the difficulty of estimating future trends for very recent years, including the effect of the COVID-19 pandemic, which we have addressed below. Addressing these topics around accession and databasing time lags, we have added the following paragraph from L587:

“A significant data limitation and analytical challenge is the absence of accession dates in GBIF records. Without these dates, we cannot fully disentangle three distinct processes: (1) actual declines in specimen collecting activity, (2) time lags between collection and local databasing, and (3) delays between local databasing and GBIF contribution. To address this data limitation, we implemented forecasting analyses using historical GBIF database snapshots. See Sections 5 and 6 of the Supplementary Information for a detailed discussion of these data challenges.”

2. Impacts of the pandemic on data collection and reporting. A major finding of this paper is a sharp decrease in GBIF specimen records post-2020. However, this coincides with the COVID-19 pandemic, which caused reduced collecting trips, and reductions in staff and other support of collections. These issues likely exacerbated backlogs (see above), and these effects may still be felt by collections managers today. In general, the impact of the pandemic felt underdiscussed throughout.

3. Considering the above, I think the main figures in the paper should be restricted to 1950-2019, with full dataset in the supplemental material. Similarly, I would like to see the forecasting model repeated using December 2019 as the final date to see if avoiding the pandemic changes any predictions for trends in collections. Alternatively, the authors may attempt to quantify the impact of backlogs by some compensatory mechanism. This could be achieved by comparing the number of specimens collected and archived in 2020 (or another recent year) to the number of specimens collected in 2020 but archived at a later date.

Thank you for your comments and suggestions regarding the role of the COVID-19 pandemic. We agree with and have implemented your suggestions to limit the year range for analysis to 1950-2019 for Figures 1 and 2. For reference, the full year range of 1800-2024 is available in Supplementary Figure 1. We have added the following paragraph in the Methods to address this at L894:

“For our primary analyses and forecasting (Figures 1 and 2), we restricted data to pre-2020 for two key reasons. First, recent collection years have inherently greater uncertainty due to a shorter elapsed time with unknown databasing and accession delays, making forecasting analyses and inferences for long-term trends based on these years less reliable. Second, the COVID-19 pandemic (2020 onwards) severely disrupted collection operations through cancelled expeditions, reduced staffing, and facility closures, exacerbating specimen backlogs. By focusing on pre-pandemic data, we avoid conflating long-term trends with these extraordinary disruptions while ensuring more robust analyses. Supplementary Figure 1 include results up to 2024 for the primary analyses.”

We also note the unknown role of the pandemic when discussing forecasting results at L328:

“Importantly, based on 5-year and 10-year ARIMA modelling results, the observed trends of long-term declines for Chordata and Plantae specimens are forecast to remain at a similar magnitude and timing. We note however, that at this stage, it is difficult to confidently forecast the transient effect of the 2020-2025 global pandemic on this trend.”

Regarding your points about repeating the forecasting model and comparison: Each ARIMA model was fit separately for each collection year, so excluding the ARIMA model for collection year 2020 does not impact the forecast inferences for collections years 2019 and prior. We agree it would be interesting to compare e.g. number of specimens collected and archived in 2020 vs those archived later – unfortunately, we do not have access to databasing/accession dates in GBIF, so we are unable to conduct such an analysis. We address this limitation in the following paragraph from L587:

“A significant data limitation and analytical challenge is the absence of accession dates in GBIF records. Without these dates, we cannot fully disentangle three distinct processes: (1) actual declines in specimen collecting activity, (2) time lags between collection and local databasing, and (3) delays between local databasing and GBIF contribution. To address this data limitation, we implemented forecasting

analyses using historical GBIF database snapshots. See Sections 5 and 6 of the Supplementary Information for a detailed discussion of these data challenges.”

4. Changes in spatial trends in collecting. While this is a very large dataset, many collections do not contribute records to GBIF. I have some concern that the reported declines in collections in some regions of the world may be due to changes in North America/Europe collections that contribute greatly to GBIF decreasing collections in other countries, while developing countries increase their collections but are less likely to contribute records to GBIF. Is there any data on the geographic distribution of collections that are contributing to GBIF? This would make a nice supplemental figure showing the locations of these collections.

Thank you for raising this important point about geographic biases in GBIF data contribution. We agree that this is a significant consideration when interpreting spatial trends in specimen collection. To address your concern, we have created a new supplementary figure (Supplementary Figure 18) showing the global distribution of institutions contributing to GBIF, based on data from the Global Registry of Scientific Collections (GRSciColl; Schindel et al., 2016). This visualisation reveals a notable concentration of contributing institutions in North America and Europe, with fewer institutions in developing regions, supporting your hypothesis that geographic biases in data contribution may influence our findings. We have added text to the spatial distribution section in the main paper to acknowledge this limitation, from L276:

“It is important to note that the geographic distribution of institutions contributing to GBIF is not uniform, with a higher concentration in North America and Europe (Supplementary Figure 18). This geographic bias in data contribution may influence the observed spatial patterns in specimen collection trends, particularly in regions with fewer contributing institutions. The extent to which these geographic biases influence observed trends in collecting is difficult to quantify.”

5. Figure 4 could use some polishing. The placement of the text was difficult to read in parts and some of the distributions of labels were unclear.

Thank you for your feedback. As multiple reviewers raised concerns with Figure 4, we have removed it from the manuscript.

6. Additional citations are needed in the “potential drivers of declines in collection data” section (Lines 282-325) to reflect recent literature on these topics.

Thank you for your feedback. We have added the following citations to support our claims in paragraphs (a) to (f) in the ‘Potential Drivers Of Decline’ section:

22. Kemp, C. The endangered dead. *Nature* **518**, 293 (2015).
23. Suarez, A. V & Tsutsui, N. D. The value of museum collections for research and society. *Bioscience* **54**, 66–74 (2004).
24. Bakker, F. T. *et al.* The Global Museum: natural history collections and the future of evolutionary science and public education. *PeerJ* **8**, e8225 (2020).

25. Carroll, S. *et al.* The CARE principles for indigenous data governance. *Data Sci J* **19**, (2020).
26. Walter, M., Kukutai, T., Carroll, S. R. & Rodriguez-Lonebear, D. *Indigenous Data Sovereignty and Policy*. (Taylor & Francis, 2021).
27. Buck, M. & Hamilton, C. The Nagoya Protocol on access to genetic resources and the fair and equitable sharing of benefits arising from their utilization to the Convention on Biological Diversity. *Review of European Community & International Environmental Law* **20**, 47–61 (2011).
28. Hedrick, B. P. *et al.* Digitization and the future of natural history collections. *Bioscience* **70**, 243–251 (2020).
29. Rocha, L. A. *et al.* Specimen collection: an essential tool. *Science (1979)* **344**, 814–815 (2014).
30. Schindel, D. E. & Cook, J. A. The next generation of natural history collections. *PLoS Biol* **16**, e2006125 (2018).
31. Funk, V. A. Collections-based science in the 21st century. *J Syst Evol* **56**, 175–193 (2018).
32. Sandall, E. L. *et al.* A globally integrated structure of taxonomy to support biodiversity science and conservation. *Trends Ecol Evol* **38**, 1143–1153 (2023).
33. Moura, M. R. & Jetz, W. Shortfalls and opportunities in terrestrial vertebrate species discovery. *Nat Ecol Evol* **5**, 631–639 (2021).

7. L385-392 – In this paragraph, you suggest that the responsibility for promoting the value of natural history collections should fall primarily to collection scientists. Given the reduction in collection scientist staff size and the growth of the value in collections in other areas, this should be the responsibility of the wider scientific community.

Thank you for this important point. We agree that the responsibility for promoting the value of natural history collections should extend beyond collection scientists to the broader scientific community, especially given the staffing constraints faced by many collections. We have revised this paragraph to emphasize the need for collaborative advocacy across disciplines, from L530:

“The scientific community as a whole needs to collaboratively advocate for the value of digitised and mobilised collections data in tackling global environmental problems. While collection scientists bring crucial expertise, the growing applications of specimen data across diverse disciplines necessitate broader engagement from researchers in ecology, genomics, climate science, and other fields. Building awareness of the tremendous value inherent in natural history collections, both in the public (e.g., via citizen science initiatives) and in funding bodies is crucial for securing resources needed to grow collections and address structural and capability constraints. Similarly, structured exploration of the data needs of relevant research disciplines and applications will help to shape what collections data should most usefully look like in the future. Collections should adopt a forward-looking mission that emphasises eco-evolutionary processes and predictive tools, while maintaining focus on fundamental species discovery in biodiversity hotspots and understudied areas.”

Reviewer #3 (Remarks on code availability):

The authors have a well-organized Zendo repository containing the code to clean and create figures.

Thank you for your time and positive feedback. We believe that our manuscript is much improved thanks to your thorough and helpful comments.

Reviewer #4 (Remarks to the Author):

Thank you for your time and positive feedback. We believe that our manuscript is much improved thanks to your thorough and helpful comments.